

# 1 Characteristics and mixing state of amine-containing

# 2 particles at a rural site in the Pearl River Delta, China.

Chunlei Cheng[1,2], Zuzhao Huang[3], Chak K. Chan[4], Yangxi Chu[4], Mei Li[1,2]*, Tao
Zhang[5], Yubo Ou[5], Duohong Chen[5], Peng Cheng[1,2], Lei Li[1,2], Wei Gao[1,2], Zhengxu
Huang[1,2], Bo Huang[1,2,6], Zhong Fu[6], Zhen Zhou[1,2]*
[1]Institute of Mass Spectrometer and Atmospheric Environment, Jinan University, Guangzhou 510632,
China
[2]Guangdong Provincial Engineering Research Center for on-line source apportionment system of air po
llution, Guangzhou 510632, China
[3]Guangzhou Environmental Technology Assessment Center, Guangzhou 510045, China
[4]School of Energy and Environment, City University of Hong Kong, Hong Kong, China
[5]State Environmental Protection Key Laboratory of Regional Air Quality Monitoring, Guangdong
Environmental Monitoring Center, Guangzhou 510308, China
[6]Guangzhou Hexin Analytical Instrument Limited Company, Guangzhou 510530, China
*Correspondence to: Mei Li (limei2007@163.com) and Zhen Zhou (zhouzhen@gig.ac.cn)
Tel: 86-20-85225991, Fax: 86-20-85225991





**Abstract.** Particulate amines play an important role for the particle acidity and
hygroscopicity and also contribute to secondary organic aerosol mass. We investigated
the sources and mixing states of particulate amines using a single-particle aerosol
mass spectrometer (SPAMS) during summer and winter 2014 at a rural site in the
Pearl River Delta, China. Amine-containing particles accounted for 12.8 % and 9.2 %
of the total detected individual particles in summer and winter, respectively.
Amine-containing particles were classified into three types: elemental and organic
carbon (ECOC), biomass burning (BB), and nitrate-rich. ECOC amine-containing
particles were the most abundant, constituting 67.2 % and 74.8 % of amine particles
in summer and winter, respectively. Both ECOC and BB type amine-containing
particles contained abundant carbonaceous and carbon-nitrogen species, as well as
sulfate and nitrate, in summer and winter. The nitrate-rich amine-containing particles
were mixed with abundant sea-salt markers in summer, indicating a possible
association between the amine emission source and marine phytoplankton. In summer,
only 6.7 % of the total amine-containing particles were found to be mixed with
ammonium, while in winter this percentage increased to 55 %. The ammonium-poor
state of amine-containing particles in summer may have been caused by the
displacement of particle-phase ammonium by amine uptake, which was more efficient
in summer at higher ambient RH (72 $\pm$ 13 %) than in winter (63 $\pm$ 11 %). In
ECOC-type amine-containing particles, the time series of the amine peak area and the
sum of the nitrate and sulfate peak areas were similar in both summer and winter,
suggesting the formation of aminium sulfate and nitrate salts. The particle acidity of
ECOC-type amine-containing particles was represented by the relative acidity ratio
($R_a$), which was defined as the ratio of the total sulfate and nitrate peak areas to the
ammonium peak area. The $R_a$ decreased from 348 $\pm$ 335 and 28 $\pm$ 14 to 10 $\pm$ 5 and 9 $\pm$
2 in summer and winter, respectively, after including amines along with the
ammonium in the acidity calculation, suggesting that it is reasonable to consider
amines when estimating particle acidity. Based on the influence of amines on
particulate ammonium and particle acidity, particulate amines could have an impact
on the newly found 'missing' source of sulfate produced from the oxidation of $SO_2$ by



NO$_2$ with NH$_3$ neutralization during haze episodes under high ambient relative
humidity in northern China.
**Keywords**: Amine; Single particles; Mixing state; Aminium salts; Particle acidity;
SPAMS.

## 77 1 Introduction

Amines, a group of nitrogen-containing organic compounds, are ubiquitous in
the atmospheric gas and particle phases (Ge et al., 2011a). A variety of low molecular
weight (LMW) aliphatic amines have been detected in emissions from anthropogenic
and natural sources, including animal husbandry, biomass burning, industrial
emissions, vehicle exhaust, and marine sources (Rogge et al., 1994; Rappert and
Muller, 2005; Calderon et al., 2007; Ngwabie et al., 2007; Ge et al., 2011a). LMW
aliphatic amines have gas-phase concentrations two orders of magnitude lower than
that of ammonia (NH$_3$) (Sorooshian et al., 2008), but are more alkaline than NH$_3$ (Ge
et al., 2011b). Due to their strong basicity and water solubility, LMW amines can
undergo acid-base reactions with sulfuric and nitric acid to form aminium salts
(Angelino et al., 2001; Sorooshian et al., 2007; Pratt et al., 2009), which has been
found to enhance new particle formation beyond the amounts produced from reactions
between acids and NH$_3$ alone (Kurten et al., 2008; Berndt et al., 2010; Place et al.,
2010; Wang et al., 2010). In addition, once partitioned into the particle phase, these
LMW aliphatic amines can enhance aerosol particle hygroscopicity (Chu et al., 2015;
Sauerwein et al., 2015). Furthermore, amines can be oxidized by OH radicals, NO$_3$
radicals, and O$_3$ in the atmosphere to form semi-volatile and non-volatile compounds,
some of which are highly toxic (Lee and Wexler, 2013), and which contribute to
secondary organic aerosol (SOA) mass (Murphy et al., 2007; Malloy et al., 2009).
The mass concentration and temporal distribution of LMW aliphatic amines in
aerosols have been studied extensively in a variety of environments, and LMW
aliphatic amines account for 2–12 % of organic mass (Day et al., 2009; Gilardoni et
al., 2009; Liu et al., 2009; Russell et al., 2009; Williams et al., 2010). In recent years,
real-time single particle mass spectrometry has been used to measure the size and





chemical composition of individual amine-containing particles with high time resolution. The mixing state and single-particle characteristics of amines have been investigated in laboratory and field environments (Moffet et al., 2008; Silva et al., 2008; Pratt et al., 2009; Huang et al., 2012; Zhang et al., 2012). Pratt et al. (2009) studied seasonal differences in aminium and ammonium salts on a single-particle basis using an aerosol time-of-flight mass spectrometer (ATOFMS) coupled with a thermodenuder and reported that the gas-to-particle partitioning of amines is dependent on particle acidity. Healy et al. (2015) investigated the temporal distributions of alkylamines at five European sites, and found that alkylamines were internally mixed with both sulfate and nitrate, which suggests that the formation of aminium salts was important at all sites. Huang et al. (2012) determined the mixing state of amine-containing particles in Shanghai and found higher number concentrations of amine-containing particles in winter than in summer, which they attributed to effective acid-base reactions between sulfuric acid and amines under low-temperature, high-RH conditions. Zhang et al. (2012) measured trimethylamine-containing particles in Guangzhou and found preferential trimethylamine gas-to-particle partitioning during fog events. These field observations emphasize the important role of acid-base reactions in the partitioning of amines from the gas phase to the particle phase. Recent laboratory studies have revealed that the exchange between amine gases and particulate $NH_3$ and/or ammonium also contributes substantially to amine content and results in a depletion of $NH_3$ and/or ammonium in the particle phase (Lloyd et al., 2009; Bzdek et al., 2010; Qiu et al., 2011; Liu et al., 2012; Chan and Chan, 2013; Chu and Chan, 2016, 2017; Sauerwein and Chan, 2017); however, the significance of such exchange reactions in the ambient environment has not been fully explored. Therefore, the influence of ammonia and particle acidity on the distribution of amines in the particle phase should be studied comprehensively through field measurements.

The aim of this study was to investigate the mixing state of a series of LMW aliphatic amines with sulfate, nitrate, and ammonium in individual particles using a single-particle aerosol mass spectrometer (SPAMS) at a rural site in the Pearl River



Delta, China. In order to explore amine origins and gas-to-particle partitioning
processes, amine-containing particles from both summer and winter were classified
into three types based on mass spectral patterns. The aminium sulfate and nitrate salt
formation processes and internal mixing state with ammonium were used to deduce
the relationship between amines and ammonium in the particle phase and the
influence of amines on particle acidity.

## 2 Methods

### 2.1 Aerosol sampling

Ambient single particles were collected and analyzed using a SPAMS at the
Guangdong Atmospheric Supersite (22.73° N, 112.93° E), a rural site in Heshan City
in the Pearl River Delta (PRD), China (Figure S1). The sampling site is surrounded by
villages and experiences little influence from local industrial emissions (Cheng et al.,
2017). The SPAMS was installed at the top of the main building, and aerosols were
introduced to the SPAMS through a 2.5 m copper tube. SPAMS sampling was
conducted continuously from 18 July to 1 August 2014 and from 27 January to 8
February 2015; several hours of data are missing due to technical maintenance.
During the sampling period, hourly $O_3$ concentrations were measured using an $O_3$
analyzer (model 49i, Thermo Scientific). Meteorological data, including temperature,
relative humidity, wind speed, and wind direction, were also measured during SPAMS
sampling.

### 2.2 SPAMS

SPAMS was designed by the Guangzhou Hexin Analytical Company based on
preexisting ATOFMS principles (Prather et al., 1994; Noble and Prather, 1996). The
setup and design of the SPAMS has been detailed previously (Li et al., 2011). Briefly,
single particles are sampled through an 80 μm critical orifice into the aerodynamic
lens at a flow rate of 75 ml min$^{-1}$. Then, the particles pass consecutively through two
laser beams (diode Nd:YAG, 532 nm) spaced 6 cm apart, and the aerodynamic
diameter of the single particle is calculated using the particle flight time and velocity
between the two laser beams. The single particle velocity is also used to calculate the





precise time at which to fire the desorption and ionization laser (Nd:YAG laser,
266nm), which is positioned 12 cm downstream from the second laser beam. After
ionization, the positive and negative ions are detected by a Z-shaped bipolar
time-of-flight mass spectrometer. In this work, the ionization laser pulse energy was
0.6 mJ and the power density was $1.06 \times 10^8$ W cm$^{-2}$ throughout the campaign. The
size range of single particles detected by SPAMS ranged from 0.2 to 2 μm, calibrated
with standard polystyrene latex spheres (Nanosphere size standards, Duke Scientific
Corp., Palo Alto) of 0.22–2.0 μm diameter before and after the campaign (Cheng et al.,

169 2017).

**2.3 Data analysis**
Particle size and chemical composition were obtained via SPAMS mass spectral
analysis using the Computational Continuation Core (COCO; version 3.0) toolkit in
Matlab. Based on previous studies using ATOFMS and SPAMS instruments
(Angelino et al., 2001; Huang et al., 2012; Zhang et al., 2012; Healy et al., 2015),
amine-containing particles were characterized by ionic markers, including $m/z$s 46
$[(CH_3)_2NH_2]^+$, 59 $[(CH_3)_3N]^+$, 74 $[(C_2H_5)_2NH_2]^+$, 86 $[(C_2H_5)_2NCH_2]^+$ or
$[C_3H_7NHC_2H_4]^+$, 101 $[(C_2H_5)_3N]^+$, 102 $[(C_3H_7)_2NH_2]^+$, 114 $[(C_3H_7)_2NCH_2]^+$, and 143
$[(C_3H_7)_3N]^+$, which correspond to dimethylamine (DMA), trimethylamine (TMA),
diethylamine (DEA), triethylamine (TEA), dipropylamine (DPA), and tripropylamine
(TPA). In this work, a particle was identified as amine-containing if it contained any
of the marker ions listed above with a relative peak area (defined as the percentage
contribution of the target ion peak area to the sum of all ion peak areas) greater than
1 %. According to this criterion, 66 331 and 70 648 amine-containing particles were
identified in summer and winter, respectively, which accounted for 12.8 % and 9.2 %
of the total detected particles. These number fractions are consistent with previously
reported observations in the PRD (Zhang et al., 2012). However, due to the absence of
fog events during the campaign, no dramatic increases in amine-containing particles
associated with high RH conditions (RH > 90 %) were observed. Amine-containing
particles were subsequently clustered using the adaptive resonance theory (ART-2a)
neural network algorithm with a vigilance factor of 0.75, a learning rate of 0.05, and a



maximum of 20 iterations.
The amine-containing particles were classified into three types: elemental and
organic carbon (ECOC), biomass burning (BB), and nitrate-rich. The ion markers and
selective criterion for these three particle types are as follows (Table S1): (1)
ECOC-type particles contain abundant carbon clusters of $m/z$s $\pm 12$ $[C]^{+/-}$, $\pm 24$ $[C_2]^{+/-}$,
$\pm 36$ $[C_3]^{+/-}$, and hydrocarbon clusters at $m/z$s 37 $[C_3H]^+$ and 43 $[C_3H_7]^+/[C_2H_3O]^+$
with relative peak areas higher than 0.5 %; (2) BB-type particles consist of any
remaining particles containing abundant signal at $m/z$ 39 $[K]^+$ (relative peak area >
30 %) and $m/z$s -59 $[C_2H_3O_2]^-$ and -73 $[C_3H_5O_2]^-$ (relative area of both peaks > 0.5 %);
(3) any remaining particles containing abundant signal at $m/z$s -46 $[NO_2]^-$ and -62
$[NO_3]^-$ with relative peak areas higher than 10% are classified as nitrate-rich. The
above classification protocol for amine-containing particles has been used in other
studies (Bi et al., 2011; Pratt et al., 2011; Zhang et al., 2013). These three types of
amine-containing particles constitute 93.5 % and 94.8 % of the total amine-containing
particles in summer and winter, respectively.
**3 Results and Discussion**
**3.1 Seasonal variation of amine-containing particles**
Meteorological conditions, namely wind speed and wind direction, are shown
during the sampling period in Figure 1. In summer, high amine-containing particle
number concentrations were associated with southwesterly and southeasterly winds at
speeds of 3–5 m s$^{-1}$, suggesting that the majority of amine-containing particles came
from regional transport. However, in winter, large amounts of amine-containing
particles were associated with northwesterly winds at speeds of 0.5–2 m s$^{-1}$, indicating
that amine-containing particles were related primarily with local emissions, such as
animal husbandry, biomass burning, and vehicle exhaust. Anthropogenic emissions
from Foshan and Guangzhou may also have contributed, as the sampling site is only
40 km and 56 km from these cities, respectively (Figure S1).
Temporal variations in amine-containing particles and meteorological data (i.e.,
RH, temperature, wind speed, and wind direction) are shown in Figure 2.





Amine-containing particles showed different trends in summer and winter, and high
concentrations of amine-containing particles were found from 22 to 24 July (in
summer) and from 5 to 8 February (in winter). The amine-containing particle count
observed in summer (66 331) was lower than it observed in winter (70 648), but the
abundance of amine-containing particles relative to the total particle count was higher
in summer (12.8 %) than in winter (9.2 %). Amine-containing particles had similar
diurnal patterns in summer and winter (Figure 3), and both showed higher count at
night; the small increase from 6:00 to 9:00 LST throughout the campaign may have
been due to local emissions from vehicle exhaust (Cadle and Mulawa, 1980). Many
field studies have revealed a strong correlation between relative humidity (RH) and
particulate amines, suggesting that high RH in fog events is favorable for the
gas-to-particle partitioning of amines (Jeong et al., 2011; Rehbein et al., 2011; Huang
et al., 2012; Zhang et al., 2012). In this work, the correlation between
amine-containing particle count and ambient RH was not obvious in either summer or
winter (Figure S2). Other factors, such as particle acidity, may have contributed to the
acid-base reactions that formed the aminium salts (Murphy et al., 2007; Kurten et al.,
2008; Silva et al., 2008).
**3.2 Mass spectra of amine-containing particles**
Amine-containing particles were categorized as ECOC, BB, and nitrate-rich both
in summer and winter. ECOC amine-containing particles were dominant, accounting
for 67.2 % and 74.8 % of particle count in summer and winter, respectively (Table 1).
ECOC particles exhibited variations similar to those in total amine-containing
particles in both summer and winter. Nitrate-rich particles were the second most
abundant in summer, during which time they accounted for 13.4 % of total
amine-containing particles, while BB particles were the second most abundant in
winter, accounting for 16.3 % of total amine-containing particles. Nitrate-rich
particles were three times more abundant in summer than in winter, suggesting the
existence of aminium nitrate salts in summer.
The average mass spectra of ECOC, BB, and nitrate-rich amine-containing
particles in summer and winter are shown in Figure 4. The ECOC amine-containing





particles in both summer and winter were characterized by high fractions of 39 [K]$^+$;
carbonaceous marker ions, including $m/z$s 27 [C$_2$H$_3$]$^+$, 29 [C$_2$H$_5$]$^+$, 36 [C$_3$]$^+$, 37 [C$_3$H]$^+$,
[C$_2$H$_3$O]$^+$, 48 [C$_4$]$^+$, 51 [C$_4$H$_3$]$^+$, 53 [C$_4$H$_5$]$^+$, 60 [C$_5$]$^+$, 63 [C$_5$H$_3$]$^+$, 65 [C$_5$H$_5$]$^+$, and
77 [C$_6$H$_5$]$^+$; and amine fragment ions at $m/z$s 46 [(CH$_3$)$_2$NH$_2$]$^+$, 59[(CH$_3$)$_3$N]$^+$, 74
[(C$_2$H$_5$)$_2$NH$_2$]$^+$, and 86 [(C$_2$H$_5$)$_2$NCH$_2$]$^+$/[C$_3$H$_7$NHC$_2$H$_4$]$^+$ in the positive mass
spectrum. The ECOC particle negative mass spectrum was characterized by strong
carbon-nitrogen fragment signals at $m/z$s -26 [CN]$^-$ and -42 [CNO]$^-$, as well as
abundant secondary ions at $m/z$s -46 [NO$_2$]$^-$, -62 [NO$_3$]$^-$, -80 [SO$_3$]$^-$, and -97 [HSO$_4$]$^-$
in both summer and winter. In many field studies, aged carbonaceous particles always
contain abundant secondary sulfate, nitrate, and ammonium ions. Interestingly, in this
work, the ammonium signal ($m/z$ 18 [NH$_4$]$^+$) was not found in ECOC
amine-containing particles in summer, and only a very small ammonium peak was
detected in winter. The low ammonium signal in ECOC amine-containing particles
may have been due to the exchange of particulate ammonium for gas-phase amines
(Lloyd et al., 2009; Qiu et al., 2011; Chan and Chan, 2012; Chan and Chan, 2013;
Chu and Chan, 2016, 2017; Sauerwein and Chan, 2017). In both summer and winter,
the mass spectra of BB amine-containing particles showed carbonaceous markers and
secondary ions similar to those found in ECOC amine-containing particles, with
additional distinct ion peaks at $m/z$ 23 [Na]$^+$ and BB markers at $m/z$s -59 [C$_2$H$_3$O$_2$]$^-$,
-71 [C$_3$H$_3$O$_2$]$^-$, and -73 [C$_3$H$_5$O$_2$]$^-$. No ammonium was found in BB amine-containing
particles in either summer or winter, likely because of exchange reactions similar to
those inferred in the ECOC amine-containing particles (see Section 3.3).

272        The nitrate-rich amine-containing particles exhibited spectral features different

from those observed in the ECOC and BB spectra. Only a few carbonaceous
fragments were observed. In summer, the nitrate-rich amine-containing particles
contained abundant sea-salt markers such as $m/z$s 23 [Na]$^+$, 62 [Na$_2$O]$^+$, and
63[Na$_2$OH]$^+$ in the positive mass spectrum and $m/z$s -93 [NaCl$_2$]$^-$ and -147
[Na(NO$_3$)$_2$]$^-$ in the negative mass spectrum. Chloride signal was not detected due to
the depletion of chloride and enrichment of nitrate in the sea-salt particle aging
process (Gard et al., 1998). In summer, 48-h backward trajectories showed that 60 %





of air masses arose from marine areas (Figure S3) and were partly associated with
marine aerosols. A small peak of *m/z* 46 [(CH$_3$)$_2$NH$_2$]$^+$ was found in the nitrate-rich
amine-containing particle spectra in summer, which likely arose from DMA produced
by marine phytoplankton (Facchini et al., 2008). The backward trajectories and mass
spectra of the nitrate-rich particles indicate that marine sources may contribute to the
amine distribution in the PRD region during summer, although the amine-containing
particles appeared to have been aged during transportation. In winter, air masses were
transported largely from urban areas like Guangzhou and Foshan (Figure S3) and
brought more anthropogenic pollutants to the sampling site. Hence, the sea-salt
markers at *m/z*s -93 [NaCl$_2$]$^-$ and -147 [Na(NO$_3$)$_2$]$^-$ were not observed in the winter
negative mass spectrum. Instead, the *m/z* 56 [Fe]$^+$ ion was identified, and, because no
dust source marker ion signals (such as Ca$^+$, CaO$^+$, and SiO$_3^-$) were found, we
speculate that iron arose mainly from industrial emissions. The nitrate-rich
amine-containing particles may have resulted from direct industrial emissions or
reactions between gaseous amines and particles from industrial emissions. Lastly, the
observed nitric acid signal (*m/z* -125 [HNO$_3$NO$_3$]$^-$) indicated strong particle acidity in
the nitrate-rich amine-containing particles in winter.
Size-resolved number distributions are shown in Figure 5 for the three types of
amine-containing particles. Both ECOC and BB amine-containing particles exhibited
unimodal distributions in the submicron mode and had a broad distribution from 0.4
to 1.0 μm in both summer and winter, which may have resulted from amine
condensation on and reaction with fine mode particles from anthropogenic emissions.
Interestingly, in summer, 37 % of the nitrate-rich amine-containing particles were
submicron in size, while 63 % were supermicron; this is reasonable, as the majority of
nitrate-rich amine-containing particles were associated with sea-salt particles from
marine sources. Healy et al. (2015) also reported large amounts of supermicron
amine-containing particles internally mixed with sea-salt particles on the island of
Corsica, France.
**3.3 Mixing states and formation processes of amine-containing particles**
To investigate the seasonal mixing states of amines with sulfate, nitrate, and





ammonium (SNA), the relative abundances of SNA-containing amine particles are
shown in Figure 6. The color scale represents the percentage contribution of
SNA-containing amine particles to total amine particles. ECOC and BB
amine-containing particles were both found to be internally mixed with sulfate
throughout the sampling period. Only a small percentage of nitrate-rich
amine-containing particles were mixed with sulfate during summer, but this
percentage increased to 53 % during winter. Amine particles containing nitrate
accounted for 39 % and 59 % of the ECOC and BB particles in summer, respectively,
and 68 % and 79 % in winter. The internal mixing state of sulfate and nitrate with
amines in single particles suggests the possible formation of aminium sulfate and
nitrate salts. Only 6.7 % of the total amine-containing particles contained ammonium
in summer, while percentage increased dramatically to 55 % in winter, indicating an
ammonium-poor state in summer and an ammonium-rich state in winter. The
particle-phase mixing states of ammonium and amines may be influenced by seasonal
changes in ambient meteorological conditions (such as RH and temperature) and
chemical processes involving particle acidity and/or ammonium–amine exchange
reactions. Since no correlation was observed between RH and amine-containing
particles during the sampling period (Figure S2), seasonal variations in temperature
may have had an impact on the mixing states of ammonium and amines in the particle
phase. The temperature dropped by 15 ℃ between summer (29 $\pm$ 3.0 ℃) and winter
(14 $\pm$ 3.1 ℃). Although lower temperatures favor the partitioning of both gaseous
ammonia and amines into the particulate phase (Huang et al., 2012), no obvious
enhancement was found in the amine-containing particle count in winter (Table 1);
this suggests that the increase in ammonium-containing amine particles was caused by
other factors, such as particle acidity and ammonium–amine exchange.
Strong sulfate and nitrate signals were detected in the amine-containing particle
mass spectra in both summer and winter, suggesting that gas-to-particle amine
partitioning may have been correlated with particle acidity. Due to the dominance of
ECOC amine-containing particles throughout the sampling period, the temporal
variations in the peak areas of amines, ammonium, and the sum of sulfate and nitrate



in ECOC particles are shown in Figure 7. The peak areas of amines and the sum of
nitrate and sulfate in ECOC particles varied similarly in summer and winter,
indicating the formation of aminium salts. Little ammonium was found in the
amine-containing particles in summer, while ammonium exhibited peak areas
comparable to those for amines in winter and temporal trends of ammonium and
amines were also similar. The sum of the sulfate and nitrate peak areas had a higher
increase rate than the amine peak area from 6 to 8 February, which may have been
caused by an increase in ammonium during this period.
Particle acidity affects the partitioning of gaseous ammonia and amines into the
particle phase and may be an important factor in the seasonal differences in
ammonium in amine-containing particles. In this study, the amine-containing particle
acidity is represented by the relative acidity ratio ($R_a$), which is defined as the ratio of
the sum of the sulfate and nitrate peak areas divided by the ammonium peak area
(Denkenberger et al., 2007; Pratt et al., 2009; Cheng et al., 2017). The ECOC particle
$R_a$ was $348 \pm 335$ in summer and $28 \pm 14$ in winter (Figure 7), indicating that the
amine-containing particles were more acidic in summer than in winter. Although high
acidity is favorable for gaseous ammonia partitioning, extremely low ammonium peak
areas were found for the amine-containing particles in summer (Figure 7), which may
have been caused by the displacement of ammonium in the particle phase by amines
to form aminium sulfate and nitrate salts; this displacement depends on RH and the
phase of the ammonium salts (Chan and Chan, 2013; Chu and Chan, 2016). The
ambient RH in summer ($72 \pm 13$ %) was higher than that in winter ($63 \pm 11$ %). Thus,
it is feasible that particles contained more water and a larger fraction of aqueous
ammonium salts in summer than in winter, which facilitated the displacement of
ammonium by amines, decreasing the ammonium concentration in the particle phase.
Particle-phase organics other than amines and aminium salts also affect amine–
ammonia exchange (Chu and Chan, 2016, 2017). However, because the SPAMS alone
cannot provide quantitative data on particle-phase organics, this issue will be
discussed in a subsequent study.
The strong correlation between the amine peak areas and the sulfate and nitrate



peak areas may indicate a feedback between amines and particle acidity. If one
includes amines in the $R_a$ calculation, the new $R_a'$ values (redefined as the ratio of the
sum of the sulfate and nitrate peak areas to the sum of the ammonium and amine peak
areas) decrease to $10 \pm 5$ and $9 \pm 2$ in summer and winter, respectively, which are 30
and 3 times lower than $R_a$ values excluding amines. In addition, the presence of
aminium salts affects the water activities and osmotic coefficients of aqueous
solutions, which may influence the calculation of particle acidity using aerosol
thermodynamic models (Sauerwein et al., 2015). Furthermore, it should be noted that
the measured pH of bulk ambient aerosols may not be representative of the actual
single particle acidity. Hence, the mixing state of aerosols should be considered in
order to comprehensively estimate the aerosol pH (Pratt et al., 2009).

Several recent studies have reported a 'missing' source of sulfate produced from

the oxidation of $SO_2$ by $NO_2$ during haze episodes with high ambient relative
humidity in northern China, and the neutralization of particulate ammonium is a key
factor in this formation mechanism (Cheng et al., 2016; Wang et al., 2016). Our study
reveals that amines significantly influence particulate ammonium and particle acidity;
thus, particulate amines could also impact this sulfate formation process during haze
episodes. In order to discuss the potential role of amines in this sulfate formation
pathway, real-time concentrations of amines, ammonium, sulfate, nitrate, and their
precursors must be available.

### 4 Summary and Conclusions

Amine-containing particles were investigated using a single particle aerosol mass

spectrometer from 18 July to 1 August 2014, and from 27 January to 8 February 2015
in Heshan, China. Amine-containing particles accounted for 12.8 % and 9.2 % of the
total detected single particles in summer and winter, respectively; both seasons were
dominated by ECOC-type amine particles at percentages of 67.2 % and 74.8 %,
respectively. The ECOC and BB amine-containing particles showed strong
carbonaceous ion, carbon-nitrogen ion, sulfate, and nitrate signals in summer and
winter. However, little ammonium was found in these amine-containing particles in



summer, and small ammonium peaks were observed in winter. In summer, the nitrate-rich amine-containing particles were mixed with abundant sea-salt markers, indicating an association between amines and marine phytoplankton emissions. In ECOC amine-containing particles, the amine peak area and the sum of the nitrate and sulfate peak areas varied similarly in summer and winter, suggesting the formation of aminium sulfate and nitrate salts. An analysis of the relative acidity ratio indicated that ECOC amine-containing particles were more acidic in summer than in winter. However, in summer, only 6.7 % of the amine-containing particles contained ammonium; this percentage increased dramatically to 55 % in winter. The ammonium-poor state of the amine-containing particles in summer may have been caused by the displacement of ammonium in the particle phase by amines to form aminium sulfate and nitrate salts. The significant influence of amines on the ratio of sulfate and nitrate to ammonium suggests that amines should be taken into consideration when estimating particle acidity.

**Author contributions:** Chunlei Cheng and Mei Li designed the experiments. Tao Zhang, Yubo Ou and Duohong Chen carried them out. Chunlei Cheng prepared the manuscript with contributions from all co-authors.

**Competing interests:** Bo Huang and Zhong Fu are both employees at Guangzhou Hexin Analytical Instrument Limited Company.

*Acknowledgements*: This work was financially supported by the National Natural Science Foundation of China (Grant No.21607056), NSFC of Guangdong Province (Grant Nos. 2017A030310180, 2015A030313339), National Key Technology R&D Program (Grant No. 2014BAC21B01), Guangdong Province Public Interest Research and Capacity Building Special Fund (Grant No. 2014B020216005), Fundamental Research Funds for the Central Universities (Grant No. 21617455), National Key Research and Development Program of China (Grant No. 2016YFC0208503), and Pearl River Nova Program of Guangzhou (No. 201506010013). Chak K. Chan would



like to acknowledge funding support from the General Fund of National Natural
Science Foundation of China (Grant No. 41675117). The authors acknowledge
sampling support from the Guangdong Atmospheric Supersite. Helpful comments and
revisions from Anthony S. Wexler and Misha I.S. Boehm are acknowledged as well.

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





**Tables and Figures**

**Table list:**

Table 1. Seasonal distributions of major types of amine-containing particles in summer and winter in the PRD, China.

**Figure captions:**

Figure 1. Seasonal distributions of amine-containing particle number concentrations associated with wind direction and wind speed in (left) summer and (right) winter.

Figure 2. Temporal variations in amine-containing particles, relative humidity (RH), temperature (T), wind speed (WS), wind direction, and three types of amine particles in Heshan, China during the entire sampling periods. Abbreviations of major particle types: elemental and organic carbon (ECOC); biomass burning (BB).

Figure 3. Diurnal variations in amine-containing particle number concentrations in summer and winter in Heshan.

Figure 4. Average ion mass spectra of ECOC, BB, and nitrate-rich amine-containing particles in (a) summer and (b) winter.

Figure 5. Size distributions of the three types of amine-containing particles in (a) summer and (b) winter.

Figure 6. Mixing states of ammonium, nitrate, and sulfate in the three types of amine-containing particles during summer and winter.

Figure 7. Temporal variations in the peak areas of amines, ammonium, and the sum of sulfate and nitrate in ECOC amine-containing particles during summer and winter. The relative acidity ratio ($R_a$), which was calculated as the ratio of the total sulfate and nitrate peak areas to the ammonium peak area, is plotted as $\log(R_a)$.



**Table:**




Table 1. Seasonal distributions of major types of amine-containing particles
in summer and winter in the PRD, China.

| Particle type | Summer (18/7-1/8, 2014) | | Winter (27/1-8/2, 2015) | |
|---|---|---|---|---|
| | Count | Percentage (%) | Count | Percentage (%) |
| ECOC | 44576 | 67.2 | 52864 | 74.8 |
| BB | 8546 | 12.9 | 11499 | 16.3 |
| Nitrate-rich | 8879 | 13.4 | 2597 | 3.7 |
| Unclassified | 4330 | 6.5 | 3688 | 5.2 |

Abbreviations of major particle types: elemental and organic carbon (ECOC), biomass burning (BB),
nitrate-rich.






**Figures:**

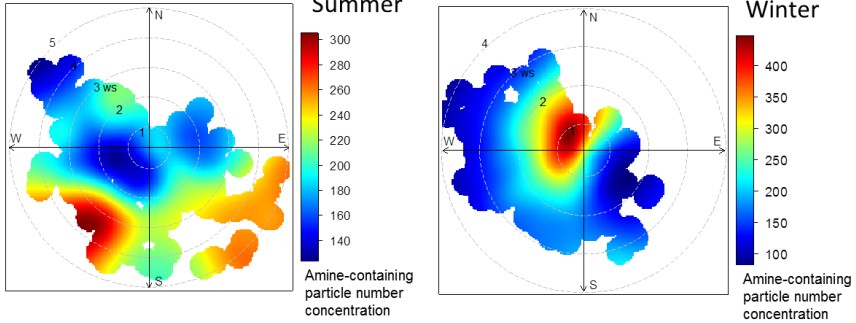



Figure 1. Seasonal distributions of amine-containing particle number concentrations
associated with wind direction and wind speed in (left) summer and (right) winter.










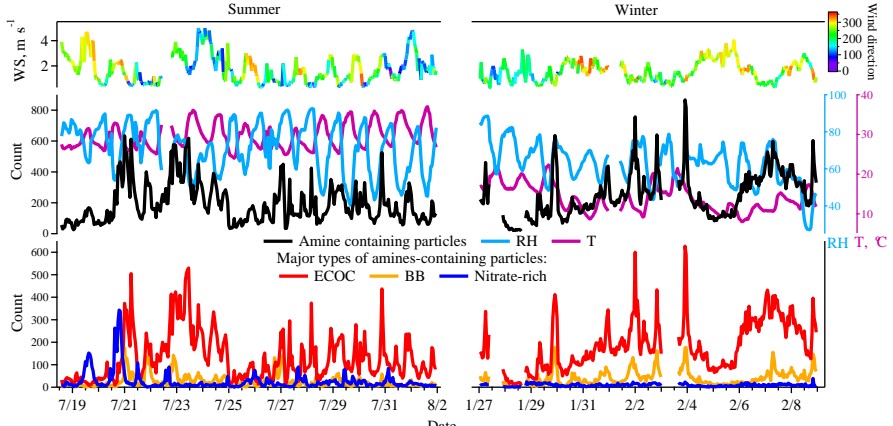


Figure 2. Temporal variations in amine-containing particles, relative humidity (RH), temperature (T), wind speed (WS), wind direction, and three types of amine particles in Heshan, China during the entire sampling periods. Abbreviations of major particle types: elemental and organic carbon (ECOC); biomass burning (BB).




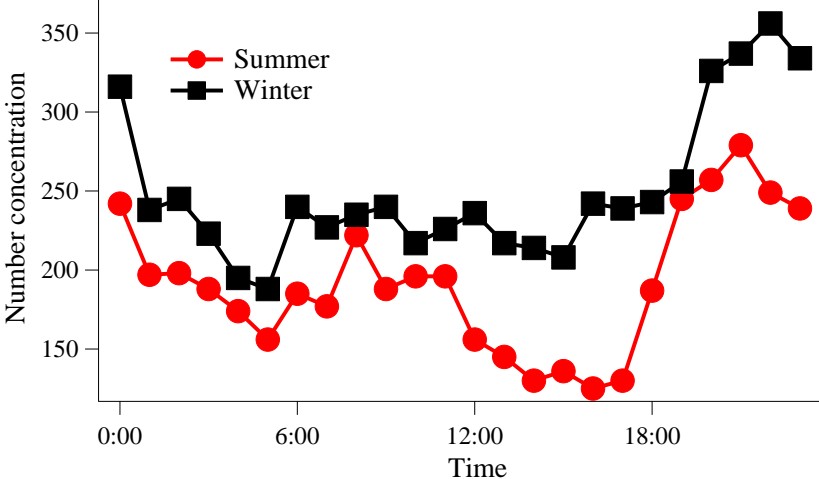


Figure 3. Diurnal variations in amine-containing particle number concentrations in summer and winter in Heshan.











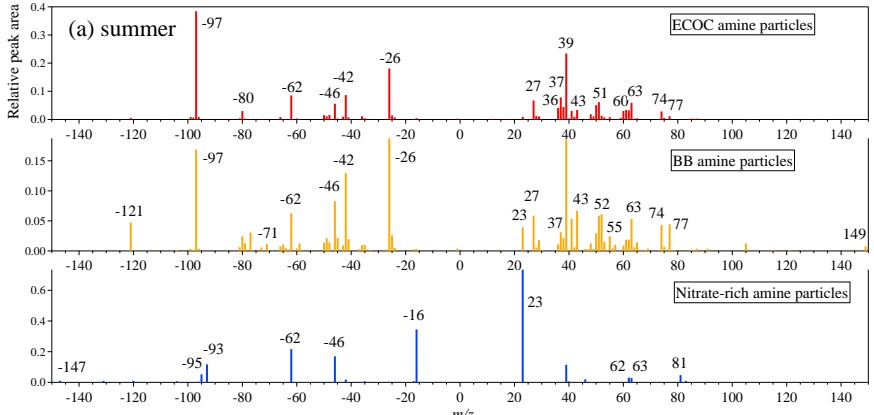


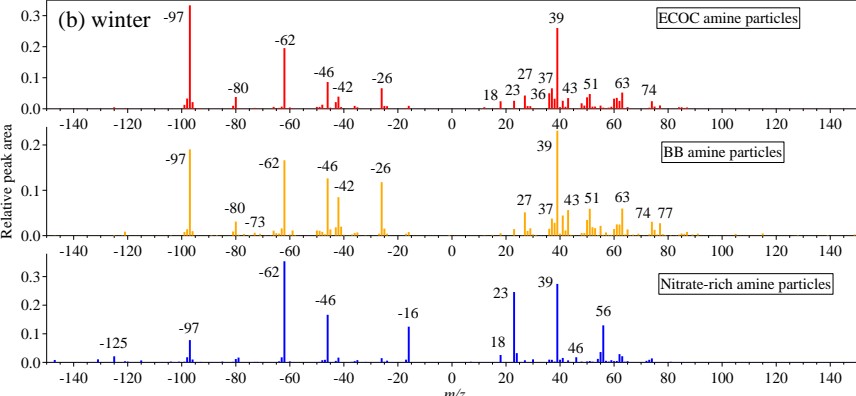


Figure 4. Average ion mass spectra of ECOC, BB, and nitrate-rich amine-containing
particles in (a) summer and (b) winter.
















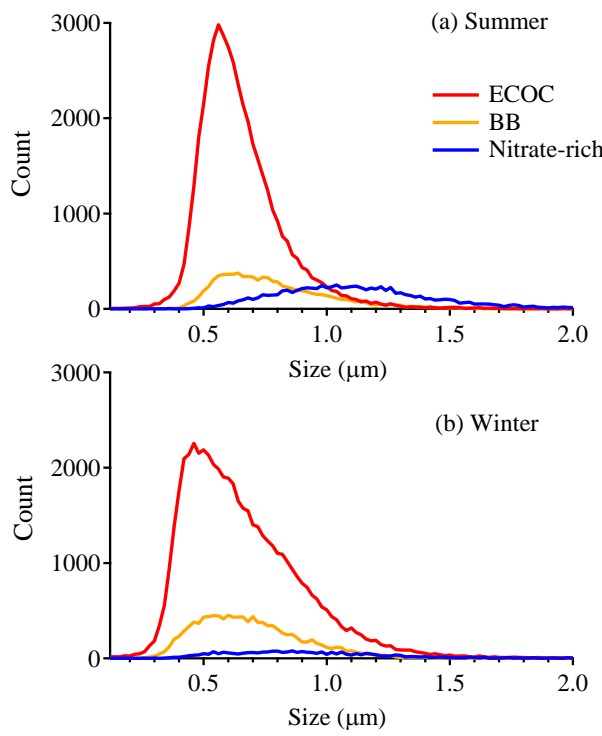


Figure 5. Size distributions of the three types of amine-containing particles in (a)
summer and (b) winter.





















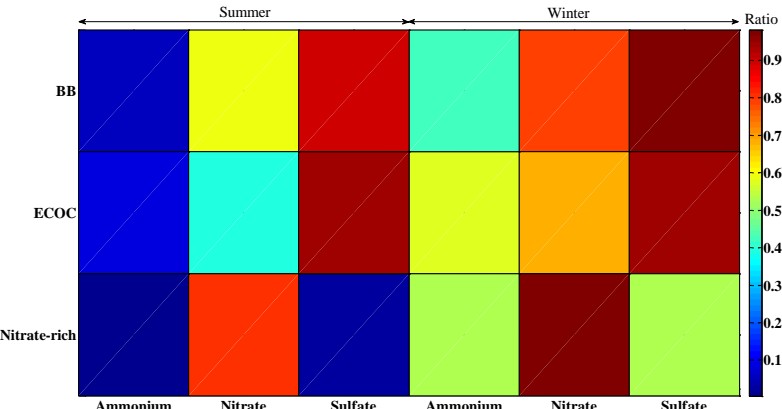


Figure 6. Mixing states of ammonium, nitrate, and sulfate in the three types of amine-containing particles during summer and winter.


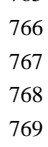

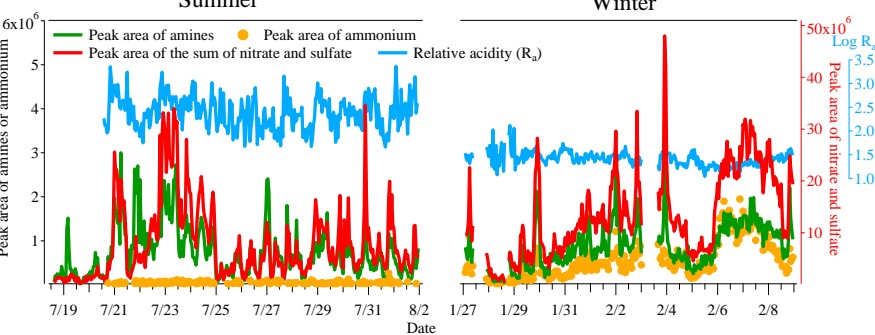


Figure 7. Temporal variations in the peak areas of amines, ammonium, and the sum of sulfate and nitrate in ECOC amine-containing particles during summer and winter. The relative acidity ratio ($R_a$), which was calculated as the ratio of the total sulfate and nitrate peak areas to the ammonium peak area, is plotted as log($R_a$).

775
776