# Peer review of "Characteristics and mixing state of amine-containing"

_Atmospheric Chemistry and Physics, 2018_

## Referee Comment (RC1) · Anonymous Referee #1 · 4 Feb 2018

This manuscript reports the mixing states of particulate amine in ambient environment using a single particle mass spectrometer. The most interesting observation made in this work is the large quantity of amine-rich particles but poor in ammonium which suggests the importance of the displacement of ammonia by amine. This observation also suggests that amines and aminiums should be considered when calculating the particle acidity. I believe this is a very important observation. The manuscript could be accepted by ACP if the authors could provide more detailed discussions to support this major conclusion. 1. Line 192-205 and Part 3.2: The mass spectral patterns of ECOC and BB particles were almost identical. The size distribution and the temporal variation were also similar. In Part 3.2, no further discussion on the differences of these two

particles types. What's the reason or necessity of this kind of classification? 2. Line 220-222: What are the different trends? High concentration amine showing on different days are not trends. 3. Line 225-234: The authors claimed that no obvious correlation between amine particle counts and RH in this work. However, both summer and winter diurnal variations showed higher counts at night. Did this diurnal pattern correlated with diurnal RH variation? 4. Line 259-263: Since this is the most important observation in this work, I strongly suggest that the author should also examine the ammonium-containing particles separately and compare with the amine-containing particles to see the differences in number fraction and temporal variation. 5. Line 283-286: More discussions should be given on the assignment of the marine source. The observed amines could be the results of secondary partitioning since the primary amines could have aged during the long-distance transport. I would suggest some detailed analysis on the other nitrate-rich particles with no sea salt mass patterns to see the differences in amine signals. 6. Line 340-342: How similar? Any correlation coefficient? 7. Line 362-364: High water content or particle acidity could also attract more ammonia to the particle phase considering the much higher concentration of ammonia in ambient environment (one or two magnitudes higher than those of amines). The authors should give more detailed discussions on the formation mechanism of these amine-rich but ammonium-poor particles. Discussion on ammonia and amine sources around sampling site is also necessary to exclude the special sources of amines. 8. Figure 1: This figure shows the spatial distributions in different seasons, not the seasonal distribution of amines.
* * *

---

## Referee Comment (RC2) · Anonymous Referee #2 · 25 Feb 2018

This paper reports on field measurements of small amines in atmospheric aerosols by single particle mass spectrometry. The collected information is used to infer the amine and particle sources and also to shed some light on the particle chemistry. The study is timely. Overall, the paper is clearly written, but several important issues need to be resolved before it can be published. I suggest a major revision.

An in-depth discussion of the three particle categories is in order. It must be told to the reader that these categories are defined operationally, based on the analysis technique. The categories do not necessarily correspond to the particle types utilized by the atmospheric aerosol community. The categories are not exclusive - a nitrate

particle may contain a strong ECOC signature, and so on. All this must be kept in mind when interpreting the particle compositions based on these three operationally defined categories.

The statement of the low ammonium ion abundance in lines 261 and 269 is in contradiction with the information given later in line 321 and also in Figure 6. This also calls for several other questions: - What is the relationship between the peak area and actual abundance of a chemical species? Is it indeed one-to-one? If not, did you perform any calibrations? Did you perform tests and calibrations for single-component particles or for mixtures? - Does the detection of ammonium depend on the particle composition/acidity? For instance, is peak area same for the particles containing same amounts of ammonium, but in the forms of ammonium nitrate, ammonium bisulfate, and ammonium sulfate? - The replacement of ammonia by amines is indeed possible. However, how realistic is it to expect that most of the ammonium will be replaced, considering that amines are an order of magnitude less atmospherically abundant than ammonia? - Similarly, how likely is it that most of chloride has been evicted from the sea-salt particles by the aging process? Could the lack of detected chloride be traced down to some other reason, such as the detection technique itself? Any calibrations with authentic chloride aerosols? What about the presence of NaCl2- clusters? Is it where all of the chloride go?

The authors must be very careful when referring to the particle mixing state. It appears that they confuse the abundances of different particles with the abundances of different chemicals in the same particle. For instance, in Line 314, do they imply that nitrate and sulfate were present in the same particles and were elevated during summer or that sulfate- and nitrate-containing particles had a high occurrence during summer? Similarly, if I am interpreting the text and Figure 6 correctly, the figure caption should not be the 'mixing state', but the fraction of particles containing different components.

As written, it appears that the authors do not treat the charge balance (ion equivalency) correctly when calculating the relative acidity. One cannot simply add the peaks of

nitrate and sulfate because the former corresponds to monoprotic and the latter to diprotic acid, respectively. Also, are peak intensities proportional to actual abundancies of chemical species?

Figure 5 is problematic. Does 'size' refer to the radius or diameter? If the y-axis is the particle count, then the plotted curves cannot be size distributions. A size distribution is expressed as dCount/dSize, but shown are apparently counts for different size bins. Those must be presented are individual points or bars, not a continuous curve. What is the bin size?

What is the shape of the particle transmission function of the aerodynamic lens? Have plots shown in Figure 5 been corrected for the size-dependent particle transmission? Frankly, I do not expect the abundance of amine-containing particles to taper off at the smaller sizes. In fact, an opposite should be true. It takes significantly less time to enrich the smaller particles with amines through the substitution reaction than to enrich the larger particles.

L175 and everywhere: replace 'm/zs' with 'm/z' L175-180: I suggest placing this information in a table L188-191: provide a reference to the processing method

---

## Referee Comment (RC3) · Anonymous Referee #3 · 26 Feb 2018

General:

This paper uses single-particle mass spectrometry to characterize particle-phase amines observed in a rural part of the Pearl River Delta in China. The analysis in this paper builds upon previous observations of amines to examine the types of amines observed and the role of acidity on the partitioning of particle phase amines using single particle mass spectrometry. Before the paper can be considered for publication, more information is needed about which amine markers were observed on the different particle types as I have reservations about the interpretation of the spectra that must be adequately addressed.

[Figure]

Major Comments:

I have several major concerns about this work that need to be addressed:

1. For the nitrate-rich particles observed in summer, m/z +46 is claimed to be an amine peak. However, several single-particle studies note the presence of m/z +46 in sea salt spectra due to the presence of 46Na2+ (e.g., Guazzotti et al., 2001; Gard et al., 1998; Gaston et al., 2011). Was m/z +46 the only "amine" ion peak noted in this particle type? If so, then this ion peak is likely a marker for sodium and not an amine. The text would need to be adjusted and conclusions about marine biogenic amines would need to be removed.

2. I have a similar concern for m/z +74 observed in biomass burning particles. This ion peak is also associated with KCl (e.g., 74KCl+) and may not be indicative of an amine. I suggest that the authors review Zauscher et al., 2013, which does show evidence of biomass burning particles containing amine markers.

3. The authors need to clarify if one amine is seen on the different particle types or multiple markers. I also suggest adding a figure showing the temporal trends for each amine marker.

4. The spectra in Figure 4 need to clearly show all of the amine markers observed on the different particle types.

Specific Comments:

Abstract:

1. It would be useful if the authors specified the percentage of amine containing particles that also contained sulfate and/or nitrate.

2. Lines 54-56: This sentence should be removed. If m/z +46 is the only marker on these aged sea salt particles, then it is likely due to 46Na2+

3. Lines 60-61: Is a 9% difference in RH enough to strongly affect the uptake of

amines? Could it be that the strength of the ammonium source has a seasonal variation?

Introduction

1. I suggest adding more references on amines and single-particle observations of amines. Please add Zauscher et al., 2013 in order to comment on biomass burning particles containing amines.

Methods

1. Lines 173-180: Which fragment goes with which amine? Many ATOFMS studies also note the presence of m/z +30, was this ion also observed? m/z +86 and m/z +118 are some of the most prevalent amines observed using ATOFMS, why wasn't m/z +118 searched for?

2. Lines 173-180: citations are needed for the ion peaks listed and the amines that they correspond to.

3. A search criterion for biomass burning aerosol was m/z -59 and m/z -73, please review Zauscher et al., 2013 for better search criteria for biomass burning aerosols.

Results

1. Figure 1 isn't very descriptive. I suggest showing this figure on a map with different sources pointed out so that the reader can see the seasonal impact of different potential sources of amines.

2. Figure 2 has too many traces. I suggest removing wind direction and better separating the amines and the RH. I also suggest showing a temporal trace for each amine marker.

3. Figure 3 needs standard error bars. Do different amine peaks show different diurnal trends?

4. Figure 4 needs amine markers clearly labeled on each particle type. I am also surprised to not see m/z +86 and m/z +118 as these are some of the most prevalent amine markers observed by single particle mass spectrometry.

5. From Figure 4, it appears as though m/z +74 is the only amine marker observed on biomass burning aerosols. m/z +74 is also a marker for KCl (e.g., 74KCl+) as noted in Zauscher et al., 2013 and may be misclassified as an amine in this work. Were any other amine markers observed on this particle type?

6. Figure 5 should clearly state that the x-axis is vacuum aerodynamic diameter.

7. Figure 6 could be a subtraction plot between the two seasons to better illustrate the seasonal difference in the mixing-state of the amine-containing particles.

8. The biomass burning spectra and ECOC spectra look the same. I think these should be the same particle type. See Ault et al., 2010 for representative ECOC spectra, which do not contain intense ions at 39K+.

9. Lines 277-279: chloride is not completely displaced and is detected as 81, 83Na2Cl+ and 93, 95NaCl2-.

10. Lines 281-283: I am not convinced that this is an observation of a marine-derived amine. The spectra clearly show aged sea salt particles, which should contain an ion peak at 46Na2+. Therefore, the peak at m/z +46 is likely not an amine.

11. Lines 333-334: could it be that the source strength of ammonium shows a seasonal cycle? This is a more likely explanation than differences in partitioning caused by a very small change in RH.

12. The authors are quite redundant about the displacement of ammonium by amines. This should be re-read and redundancies should be reduced.

Conclusions:

1. The authors should comment on which amine markers were most prevalent and

on which particle types. This would be an interesting conclusion that may also tie into different potential amine sources.

---

## Author Comment (AC1) · 22 May 2018

**Response to the comments of Anonymous Referee #1**

[Atmospheric Chemistry and Physics, MS ID: acp-2018-53]
Title: Characteristics and mixing state of amine-containing particles at a rural site in the Pearl River Delta, China.

**General comments:**
This manuscript reports the mixing states of particulate amine in ambient environment using a single particle mass spectrometer. The most interesting observation made in this work is the large quantity of amine-rich particles but poor in ammonium which suggests the importance of the displacement of ammonia by amine. This observation also suggests that amines and aminiums should be considered when calculating the particle acidity. I believe this is a very important observation. The manuscript could be accepted by ACP if the authors could provide more detailed discussions to support this major conclusion.

**Response**: Thank you for your comments. These comments are valuable and very helpful for revising and improving our paper, as well as the important guiding significance to our researches. We have studied these comments carefully and have made revisions. Our responses to the comments are itemized below.

Anything about our paper, please feel free to contact me at limei2007@163.com

Best regards!
Sincerely yours

Mei Li
May 22, 2018

**Specific comments and point by point responses:**

1. Line 192-205 and Part 3.2: The mass spectral patterns of ECOC and BB particles were almost identical. The size distribution and the temporal variation were also similar. In Part 3.2, no further discussion on the differences of these two particles types. What's the reason or necessity of this kind of classification?

**Response**: Revision made. We have combined the ECOC and BB types of particles together to investigate the mass spectra and mixing state of amine-containing particles. In addition, we have removed the nitrate-rich type particles from the manuscript due to the misinterpretation of amine marker of $^{46}(CH_3)_2NH_2^+$. Therefore, we did not rename the new group of combined ECOC and BB type particles. Actually, the amine-containing particles discussed in the revised manuscript were composed of ECOC and BB type particles. The mass spectra and size distributions of ECOC and BB type particles were replaced by total amine-containing particles, and all the discussions about ECOC and BB type particles were all replaced by total amine-containing particles.

[revised manuscript text omitted]

2. Line 220-222: What are the different trends? High concentration amine showing on different days are not trends.

**Response**: We have changed the expression to "The increase of amine-containing particles was mostly associated with high relative humidity (RH) at night in summer, while no direct connection between particle counts and RH was found in winter (Figure S2 a and b). High counts of amine-containing particles that extended in a few days were found from 22 to 24 July (in summer) and from 5 to 8 February (in winter)." in lines 241-245.

3. Line 225-234: The authors claimed that no obvious correlation between amine particle counts and RH in this work. However, both summer and winter diurnal variations showed higher counts at night. Did this diurnal pattern correlated with diurnal RH variation?

**Response**: No good correlations between diurnal amine-containing particles (1-hour average of particle count) and diurnal RH were found. We have added the correlations between diurnal amine-containing particles and diurnal RH.

"In this work, the correlation between amine-containing particle count and ambient RH was not obvious in either summer or winter (Figure S2). Other factors, such as particle acidity, may have contributed to the acid-base reactions that formed the aminium salts (Murphy et al., 2007;Kurten et al., 2008;Silva et al., 2008)." has been revised to "In this work, although the increase of amine-containing particle count mostly occurred at night, no obvious correlations between diurnal amine-containing particles and RH were found in summer ($r^2$=0.33) and winter ($r^2$=0.0003) (Figure S2). The increase of amine-containing particles at night may be

influenced by particle acidity and emission sources of amines (Murphy et al., 2007;Kurten et al., 2008;Silva et al., 2008)." in lines 263-268.

The new Figure S2:

[Figure]

Figure S2. The linear regressions between total amine-containing particle count and relative humidity (RH) in summer (a) and winter (b), and diurnal amine-containing particle count and diurnal RH in summer (c) and winter (d) in Heshan: the diurnal amine-containing particles were averaged from hourly particle counts during sampling period in summer and winter separately.

4. Line 259-263: Since this is the most important observation in this work, I strongly suggest that the author should also examine the ammonium-containing particles separately and compare with the amine-containing particles to see the differences in number fraction and temporal variation.

   **Response**: As suggested by you, the total ammonium-containing particles were

selected, which contained $^{18}NH_4^+$ with relative area larger than 1%. With this threshold, 18 336 and 235 312 of ammonium-containing particles were detected in summer and winter separately, accounting for 3.6% and 32.6% of the total detected particles. The temporal variation and mass spectra of ammonium-containing particles were discussed in the manuscript. The lower abundance of $NH_4^+$ in particulate phase may be due to the low emission sources of ammonia and preferred partitioning in gas phase in summer.

"The seasonal differences of the mixing state of amines and $NH_4^+$ may be influenced by the seasonal variation of source strength of $NH_4^+$. To investigate the temporal variation and abundance of $NH_4^+$ in total detected single particles, the total $NH_4^+$-containing particles were identified with relative area of $^{18}NH_4^+$ larger than 1%. Using this criterion, 18336 and 235312 of $NH_4^+$-containing particles were detected in summer and winter separately, accounting for 3.6% and 32.6% of the total detected particles. The averaged positive and negative ion mass spectra of $NH_4^+$-containing particles are exhibited in Figure 6. During entire sampling period the $NH_4^+$-containing particles were characterized by abundant hydrocarbon fragments and secondary organic species like $^{43}C_2H_3O^+$ and $^{89}HC_2O_4^-$, as well as strong signals of $^{26}CN^-$, $^{42}CNO^-$, $^{62}NO_3^-$ and $^{97}HSO_4^-$, indicating an aging state of $NH_4^+$-containing particles. Also, 20% of $NH_4^+$-containing particles contained $^{74}(C_2H_5)_2NH_2^+$, which indicates a close connection between $NH_3$ and diethylamine (DEA), possibly due to the similar emission sources. Temporal variations of total amine-containing particles, total ammonium-containing ($NH_4^+$-containing) particles and particles containing both ammonium and amine ($NH_4^+$-amine) are shown in Figure 7. The total $NH_4^+$-containing particles and $NH_4^+$-amine particles were both much lower in summer than in winter. This seasonal difference may be due to the low emission sources of ammonia and preferred partitioning in gas phase in summer. Backward trajectories analysis (Figure 1) showed that in summer the air mass was mainly from south of the sampling site and linked to the marine region with low emission of anthropogenic pollutants. By contrast, in winter, the air mass was mainly from northwest of the sampling site and associated with relatively polluted megacities like Guangzhou and Foshan. RH does not seem to exert a major influence on particulate $NH_4^+$ (Huang et al., 2012), because lower abundance of $NH_4^+$ was observed in summer (RH = 72 ± 13%) than in winter (RH =63 ± 11%)." has been added to the manuscript in lines 309-335.

[Figure]

Figure 6. Mass spectra of total ammonium-containing ($NH_4^+$-containing) particles in summer and winter. The color bars represent each peak area corresponding to a specific fraction in individual particles.

[Figure]

Figure 7. Temporal variations of total amine-containing particles, total ammonium-containing particles and particles containing both ammonium and amine ($NH_4^+$-amine) during sampling period in Heshan.

5. Line 283-286: More discussions should be given on the assignment of the marine source. The observed amines could be the results of secondary partitioning since the primary amines could have aged during the long-distance transport. I would suggest some detailed analysis on the other nitrate-rich particles with no sea salt mass patterns to see the differences in amine signals.

**Response**: Indeed, the amines in the nitrate-rich particles could be formed from the gas to particle partitioning process in addition to direct marine emissions. However, as pointed by the reviewer #3, m/z +46 could be $^{46}Na_2^+$ in addition to as amine marker of $^{46}(CH_3)_2NH_2^+$, because no other amine markers were found and nitrate-rich particles were enriched with sodium salts like $^{62}Na_2O^+$, $^{81}Na_2Cl^+$ and

$^{147}$Na(NO$_3$)$_2$-, so we removed nitrate-rich particles from amine-containing particles, and all the related discussions were removed too.

"In this work the m/z +46-containing particles had no other amine markers as listed above, besides, these particles were enriched with sodium salts like $^{62}$Na$_2$O$^+$, $^{81}$Na$_2$Cl$^+$ and $^{147}$Na(NO$_3$)$_2^-$. Thus, m/z +46-containing particles were not classified as amine-containing particles." has been added in the lines 216-220.

"The nitrate-rich amine-containing particles exhibited spectral features different from those observed in the ECOC and BB spectra. Only a few carbonaceous fragments were observed. In summer, the nitrate-rich amine-containing particles contained abundant sea-salt markers such as $m/z$s 23 [Na]$^+$, 62 [Na$_2$O]$^+$, and 63[Na$_2$OH]$^+$ in the positive mass spectrum and $m/z$s -93 [NaCl$_2$]$^-$ and -147 [Na(NO$_3$)$_2$]$^-$ in the negative mass spectrum. Chloride signal was not detected due to the depletion of chloride and enrichment of nitrate in the sea-salt particle aging process. In summer, 48-h backward trajectories showed that 60 % of air masses arose from marine areas (Figure S3) and were partly associated with marine aerosols. A small peak of $m/z$ 46 [(CH$_3$)$_2$NH$_2$]$^+$ was found in the nitrate-rich amine-containing particle spectra in summer, which likely arose from DMA produced by marine phytoplankton. The backward trajectories and mass spectra of the nitrate-rich particles indicate that marine sources may contribute to the amine distribution in the PRD region during summer, although the amine-containing particles appeared to have been aged during transportation. In winter, air masses were transported largely from urban areas like Guangzhou and Foshan (Figure S3) and brought more anthropogenic pollutants to the sampling site. Hence, the sea-salt markers at $m/z$s -93 [NaCl$_2$]$^-$ and -147 [Na(NO$_3$)$_2$]$^-$ were not observed in the winter negative mass spectrum. Instead, the $m/z$ 56 [Fe]$^+$ ion was identified, and, because no dust source marker ion signals (such as Ca$^+$, CaO$^+$, and SiO$_3^-$) were found, we speculate that iron arose mainly from industrial emissions. The nitrate-rich amine-containing particles may have resulted from direct industrial emissions or reactions between gaseous amines and particles from industrial emissions. Lastly, the observed nitric acid signal ($m/z$ -125 [HNO$_3$NO$_3$]$^-$) indicated strong particle acidity in the nitrate-rich amine-containing particles in winter." has been removed from the manuscript.

6. Line 340-342: How similar? Any correlation coefficient?

**Response**: Revision made. "The peak areas of amines and the sum of nitrate and sulfate in ECOC particles varied similarly in summer and winter, indicating the formation of aminium salts." have been revised to "The peak areas of amines and the sum of nitrate and sulfate had similar variation patterns both in summer and winter. The linear regression between them showed robust correlations both in summer (r$^2$=0.74) and winter (r$^2$=0.88) (Figure S3), indicating the formation of aminium salts." in lines 337-341.

[Figure]

Figure S3. The linear regression between peak area of amines and the peak area of sulfate and nitrate in summer and winter.

7. Line 362-364: High water content or particle acidity could also attract more ammonia to the particle phase considering the much higher concentration of ammonia in ambient environment (one or two magnitudes higher than those of amines). The authors should give more detailed discussions on the formation mechanism of these amine-rich but ammonium-poor particles. Discussion on ammonia and amine sources around sampling site is also necessary to exclude the special sources of amines.

**Response**: Revision made. The analysis of backward trajectories (Figure 1) showed that in summer the air mass was mainly from south of the sampling site and linked to the marine region with low emission of anthropogenic pollutants. By contrast, in winter, the air mass was mainly from northwest of the sampling site and associated with relatively polluted megacities like Guangzhou and Foshan. The low abundance of ammonium in amine-containing particles in summer is likely due to the low emission of ammonia. In addition, according to the study of Sauerwein and Chan, the co-uptake of dimethylamine (DMA) and ammonia ($NH_3$) by sulfuric acid particles at 50% RH led to particle-phase dimethylaminium ($DMAH^+$)/ammonium ($NH_4^+$) molar ratio up to four times that of gas-phase DMA/ammonia molar ratio (0.1 and 0.5), suggesting the displacement of $NH_4^+$ by DMA during the uptake process (Sauerwein and Chan, 2017). Thus, it is feasible for the displacement of $NH_4^+$ by DMA when the concentration of amines is one order of magnitude lower than $NH_3$ in gas phase. We have extended the discussions about the possible role of ammonium–amine exchange reactions in amine-containing particles in the manuscript. Besides, the influence of ambient RH and particle acidity on the gas to particle partitioning of ammonia has been discussed in the manuscript.

"Temporal variations of total amine-containing particles, total ammonium-containing ($NH_4^+$-containing) particles and particles containing both ammonium and amine ($NH_4^+$-amine) are shown in Figure 7. The total $NH_4^+$-containing particles and $NH_4^+$-amine particles were both much lower in summer than in winter. This seasonal difference may be due to the low emission sources of

ammonia and preferred partitioning in gas phase in summer. Backward trajectories analysis (Figure 1) showed that in summer the air mass was mainly from south of the sampling site and linked to the marine region with low emission of anthropogenic pollutants. By contrast, in winter, the air mass was mainly from northwest of the sampling site and associated with relatively polluted megacities like Guangzhou and Foshan. RH does not seem to exert a major influence on particulate $NH_4^+$ (Huang et al., 2012), because lower abundance of $NH_4^+$ was observed in summer (RH = 72 ± 13%) than in winter (RH =63 ±11%)." have been added and revised in lines 323-335.

"Although high acidity promotes gaseous ammonia partitioning, extremely low ammonium peak areas were found for the amine-containing particles in summer (Figure 8), which may be associated with ammonium–amine exchange reactions in addition to the low emission source of ammonia. The exchange between amine gases and particulate $NH_3$ and/or ammonium highly depends on the RH and particle acidity (Chan and Chan, 2013;Chu and Chan, 2016). According to the study of Sauerwein and Chan, the co-uptake of dimethylamine (DMA) and ammonia ($NH_3$) by sulfuric acid particles at 50% RH led to particle-phase dimethylaminium ($DMAH^+$) to ammonium ($NH_4^+$) molar ratio up to four times that of gas-phase DMA to ammonia molar ratio (0.1 and 0.5), suggesting the displacement of $NH_4^+$ by DMA during the uptake process (Sauerwein and Chan, 2017). In this work, the ambient RH and acidic particles containing abundant sulfate and nitrate were similar to the experimental conditions used in Sauerwein and Chan (2017). In summer 8% of amine-containing particles contained $NH_4^+$, while 25% of ammonium-containing particles contained amines (Figure 7). Although the gas-phase concentrations of amines and $NH_3$ are not quantified, higher abundance of amines in ammonium-containing particles than that of ammonium in amine-containing particles suggests a possible ammonium–amine exchange reactions in acidic particles in summer." have been revised in lines 356-373.

8. Figure 1: This figure shows the spatial distributions in different seasons, not the seasonal distribution of amines.

    **Response**: In order to clearly show the impact of different sources on the variation of amine-containing particles, we use the results of backward trajectories to show the connection between the abundance of amine-containing particles and the air sources. The "seasonal distributions" has been changed to "spatial distributions", and all the related revisions are as follows:

[revised manuscript text omitted]

---

## Author Comment (AC2) · 22 May 2018

**Response to the comments of Anonymous Referee #2**

[Atmospheric Chemistry and Physics, MS ID: acp-2018-53]
Title: Characteristics and mixing state of amine-containing particles at a rural site in the Pearl River Delta, China.

**General comments:**
This paper reports on field measurements of small amines in atmospheric aerosols by single particle mass spectrometry. The collected information is used to infer the amine and particle sources and also to shed some light on the particle chemistry. The study is timely. Overall, the paper is clearly written, but several important issues need to be resolved before it can be published. I suggest a major revision.

**Response**: Thank you for your comments. These comments are valuable and very helpful for revising and improving our paper, as well as the important guiding significance to our researches. We have studied these comments carefully and have made corrections. Our responses to the comments are itemized below.

Anything about our paper, please feel free to contact me at limei2007@163.com

Best regards!
Sincerely yours

Mei Li
May 22, 2018

**Specific comments and point by point responses:**

1. An in-depth discussion of the three particle categories is in order. It must be told to the reader that these categories are defined operationally, based on the analysis technique. The categories do not necessarily correspond to the particle types utilized by the atmospheric aerosol community. The categories are not exclusive - a nitrate particle may contain a strong ECOC signature, and so on. All this must be kept in mind when interpreting the particle compositions based on these three operationally defined categories.

**Response**: Indeed, the classification criterions are subjectively defined and should be described clearly in the manuscript. "It should be noted that amine-containing particles are operationally defined and not exclusive, which also contained various chemical species in addition to amines." have been added in lines 198-200. Several marker ions of BB and nitrate-rich type particles were misinterpreted during the classification, so we have removed the classification and description of nitrate-rich particles, and combined the ECOC and BB type particles together to investigate the mass spectra and mixing state of amine-containing particles. We characterized the total amine-containing particles and particles containing three major amine markers of $^{59}(CH_3)_3N^+$, $^{74}(C_2H_5)_2NH_2^+$ and $^{86}(C_2H_5)_2NCH_2^+$.

The paragraphs of particle categorization have been replaced by "Particle size and chemical composition were obtained via SPAMS mass spectral analysis using the Computational Continuation Core (COCO; version 3.0) toolkit in Matlab. According to the field studies of ATOFMS and SPAMS, it is difficult to accurately determine the number concentration of ambient particles using SPAMS alone due to the size-dependent transmission efficiencies of particles through aerodynamic lens and composition dependent matrix effect (Gross et al., 2000;Pratt and Prather, 2012). Thus, the particle counts and size distributions presented in this work should be interpreted as semi-quantitative and serve as a basis of comparison analysis (Healy et al., 2012). Based on previous studies using ATOFMS and SPAMS instruments (Angelino et al., 2001;Huang et al., 2012;Zhang et al., 2012;Zauscher et al., 2013;Healy et al., 2015), amine-containing particles were characterized by marker ions, including *m/z* $^{59}(CH_3)_3N^+$, $^{74}(C_2H_5)_2NH_2^+$, $^{86}(C_2H_5)_2NCH_2^+$, $^{101}(C_2H_5)_3N^+$, $^{102}(C_3H_7)_2NH_2^+$, and $^{143}(C_3H_7)_3N^+$ (Table 1). In this work, a particle was identified as amine-containing if it contained any of the marker ions listed above with a relative peak area (defined as the percentage contribution of the target ion peak area to the sum of all ion peak areas) greater than 1%. It should be noted that amine-containing particles are operationally defined and not exclusive, which also contained various chemical species in addition to amines. According to this criterion, 57452 and 68026 amine-containing particles were identified in summer and winter, respectively, which accounted for 11.1 % and 9.4 % of the total detected particles. These number fractions are consistent with previously reported observations in the PRD (Zhang et al., 2012). However, due to the absence of fog events during the campaign, no dramatic increases in amine-containing particles associated with high RH conditions (RH > 90 %) were observed. Marker ions of $^{59}(CH_3)_3N^+$, $^{74}(C_2H_5)_2NH_2^+$, $^{86}(C_2H_5)_2NCH_2^+$ were detected

as the most abundant amines species during the sampling period, so particles containing each marker ion were selected to investigate the possible sources and characteristics of amine-containing particles. $^{30}CH_3NH^+$ is also an amine marker which has been reported by other single particle studies (Phares et al., 2003;Glagolenko and Phares, 2004). In this work the peak intensity of $^{30}CH_3NH^+$ was much lower compared with other amine markers, and all the particles containing $^{30}CH_3NH^+$ had strong signal of $^{74}(C_2H_5)_2NH_2^+$, so the $^{30}CH_3NH^+$-containing particles were not specifically discussed. Ion of m/z +46 was detected in the ambient single particles, which could be the amine marker of $^{46}(CH_3)_2NH_2^+$ and/or $^{46}Na_2^+$ according to reported studies (Guazzotti et al., 2001;Gaston et al., 2011;Healy et al., 2015). In this work the m/z +46-containing particles had no other amine markers as listed above, besides, these particles were enriched with sodium salts like $^{62}Na_2O^+$, $^{81}Na_2Cl^+$ and $^{147}Na(NO_3)_2^-$. Thus, m/z +46-containing particles were not classified as amine-containing particles." in lines 183-220.

2. (1) The statement of the low ammonium ion abundance in lines 261 and 269 is in contradiction with the information given later in line 321 and also in Figure 6.

**Response**: We have revised the related discussions about ammonium in the manuscript. The original expressions in line 261, 269 and 321 have been rephrased. "Interestingly, in this work, the ammonium signal (*m/z* 18 $[NH_4]^+$) was not found in ECOC amine-containing particles in summer, and only a very small ammonium peak was detected in winter." and "No ammonium was found in BB amine-containing particles in either summer or winter, likely because of exchange reactions similar to those inferred in the ECOC amine-containing particles (see Section 3.3)." have been revised to "Interestingly, in this work, the signals of $^{18}NH_4^+$ were weak and only observed in less than 10% of amine-containing particles in summer, but moderate signal of $^{18}NH_4^+$ was detected in half of amine-containing particles in winter. The low $^{18}NH_4^+$ signal in amine-containing particles may have been due to the emission sources of ammonia and particle acidity, which will be discussed in Section 3.3." in lines 279-284. "Only 6.7 % of the total amine-containing particles contained ammonium in summer, while percentage increased dramatically to 55 % in winter, indicating an ammonium-poor state in summer and an ammonium-rich state in winter." has been revised to "Interestingly, only 8% of amine-containing particles mixed with ammonium ($NH_4^+$) in summer, while the percentage increased dramatically to 54 % in winter, indicating a relatively $NH_4^+$-poor state in summer and an $NH_4^+$-rich state in winter." in lines 305-308.

(2) This also calls for several other questions: - What is the relationship between the peak area and actual abundance of a chemical species? Is it indeed one-to-one? If not, did you perform any calibrations? Did you perform tests and calibrations for single-component particles or for mixtures? - Does the detection of ammonium depend on the particle composition/acidity? For instance, is peak area same for the

particles containing same amounts of ammonium, but in the forms of ammonium nitrate, ammonium bisulfate, and ammonium sulfate?

**Response**: The peak area of a chemical species provides a semi-quantitative result of its actual abundance in single particles, although in some cases comparison between peak area and concentration was poor due to matrix effect and transmission efficiency (Dall'Osto et al., 2006). After scaling/normalizing ATOFMS/SPAMS data based on particle counter such as SMPS (scanning mobility particle sizer) and aerodynamic particle sizer (APS), the quantitative agreement can be obtained between peak area and atmospheric concentration for major constituents of ambient aerosols. There is a great deal of work investigating the relationship between atmospheric concentrations and peak area/relative peak area from single particle mass spectrometer (ATOFMS and SPAMS) (Mansoori et al., 1994;Gross et al., 2000;Bhave et al., 2002;Bein et al., 2006;Dall'Osto and Harrison, 2006;Dall'Osto et al., 2006;Qin et al., 2006;Spencer and Prather, 2006;Snyder et al., 2009;Sullivan et al., 2009;Huang et al., 2013;Zauscher et al., 2013)

Most peak signals are corresponding to specific chemical species, such as elemental carbon, nitrate, sulfate, ammonium, oxalic acid, carbon-nitrogen fragments and most metals. Some metals (such as Al, Cu, Ti, V) and organic ions receive strong isobaric interferences with commonly observed organic fragments. For the controversial peak assignment, the abundances of other marker ions are often used to identify the most possible ions. For example, the ion of m/z +46 can be amine marker of $^{46}(CH_3)_2NH_2^+$ and $^{46}Na_2^+$ according to reported studies (Guazzotti et al., 2001;Gaston et al., 2011;Healy et al., 2015). In this work the m/z +46-containing particles had no other amine markers and the commonly observed organic fragments are not found, besides, these particles were enriched with sodium salts like $^{62}Na_2O^+$, $^{81}Na_2Cl^+$ and $^{147}Na(NO_3)_2^-$. Thus, m/z +46 was identified as $^{46}Na_2^+$ in this work.

In this work, no particle counter was used during the sampling campaign, so the reported number count of ambient particles were not scaled or calibrated, but the peak intensity and particle counts can still provide a semi-qualitative approach to investigate the seasonal characteristics and mixing state of amine-containing particles (Dall'Osto and Harrison, 2006;Healy et al., 2012).

The peak intensity of ammonium is largely dependent on its abundance in particles (Gross et al., 2000;Kane and Johnston, 2001;Liu et al., 2003;Wenzel et al., 2003;Lake et al., 2004). The particle acidity can impact the partitioning of ammonium from gas to particles, but has no influence on the ionization of ammonium. For the particles containing ammonium nitrate, ammonium bisulfate, and ammonium sulfate, the peak area of ammonium with same abundance will vary in a certain range due to the matrix effect; however, the good relationship between peak intensity of ammonium and actual concentration suggests a semi-quantitative approach through peak intensity to study the mixing state of ammonium with other species such as amines(Kane and Johnston, 2001).

(3) The replacement of ammonia by amines is indeed possible. However, how

realistic is it to expect that most of the ammonium will be replaced, considering that amines are an order of magnitude less atmospherically abundant than ammonia?

**Response**: Indeed, the concentration of ammonia in gas phase is two orders of magnitude higher than that of amines, so the variation of gas phase concentration of ammonia could have a substantial impact on the gas to particle partitioning into amine-containing particles. The seasonal different mixing state of amines and ammonium may be influenced by the seasonal variation of source strength of ammonium. However, according to the study of Sauerwein and Chan, the co-uptake of dimethylamine (DMA) and ammonia ($NH_3$) by sulfuric acid particles at 50% RH led to particle-phase dimethylaminium ($DMAH^+$)/ammonium ($NH_4^+$) molar ratio up to four times that of gas-phase DMA/ammonia molar ratio (0.1 and 0.5), suggesting the displacement of $NH_4^+$ by DMA during the uptake process (Sauerwein and Chan, 2017). Thus, it is feasible for the displacement of $NH_4^+$ by DMA when the concentration of amines is one order of magnitude lower than $NH_3$ in gas phase. We have extended the discussions about the possible role of ammonium–amine exchange reactions in amine-containing particles in the manuscript. Besides, the influence of ambient RH and particle acidity on the gas to particle partitioning of ammonia has been discussed in the manuscript.

Thus, we have revised the discussions about ammonium–amine exchange reactions to "Although high acidity promotes gaseous ammonia partitioning, extremely low ammonium peak areas were found for the amine-containing particles in summer (Figure 8), which may be associated with ammonium–amine exchange reactions in addition to the low emission source of ammonia. The exchange between amine gases and particulate $NH_3$ and/or ammonium highly depends on the RH and particle acidity (Chan and Chan, 2013;Chu and Chan, 2016). According to the study of Sauerwein and Chan, the co-uptake of dimethylamine (DMA) and ammonia ($NH_3$) by sulfuric acid particles at 50% RH led to particle-phase dimethylaminium ($DMAH^+$) to ammonium ($NH_4^+$) molar ratio up to four times that of gas-phase DMA to ammonia molar ratio (0.1 and 0.5), suggesting the displacement of $NH_4^+$ by DMA during the uptake process (Sauerwein and Chan, 2017). In this work, the ambient RH and acidic particles containing abundant sulfate and nitrate were similar to the experimental conditions used in Sauerwein and Chan (2017). In summer 8% of amine-containing particles contained $NH_4^+$, while 25% of ammonium-containing particles contained amines (Figure 7). Although the gas-phase concentrations of amines and $NH_3$ are not quantified, higher abundance of amines in ammonium-containing particles than that of ammonium in amine-containing particles suggests a possible ammonium–amine exchange reactions in acidic particles in summer." in lines 356-373.

(4) Similarly, how likely is it that most of chloride has been evicted from the sea-salt particles by the aging process? Could the lack of detected chloride be traced down to some other reason, such as the detection technique itself? Any calibrations with authentic chloride aerosols? What about the presence of NaCl2- clusters? Is it where all of the chloride go?

**Response**: We have reconsidered the classification criteria for the nitrate-rich particles which contained many sea-salt markers. Ion of m/z +46 could be $^{46}Na_2^+$ except as amine marker of $^{46}(CH_3)_2NH_2^+$, because no other amine markers were found and nitrate-rich particles were enriched with sodium salts like $^{62}Na_2O^+$, $^{81}Na_2Cl^+$ and $^{147}Na(NO_3)_2-$, so we removed nitrate-rich particles from amine-containing particles, and all the related discussions were removed too. The SPAMS has a good sensitivity to the detection of chloride, so the lack of chloride signal in particles was mainly due to the low abundance of chloride in particles. As we have removed the sea-salt particles from amine-containing particles, the depletion process of chloride is no longer discussed in this work.

"In this work the m/z +46-containing particles had no other amine markers as listed above, besides, these particles were enriched with sodium salts like $^{62}Na_2O^+$, $^{81}Na_2Cl^+$ and $^{147}Na(NO_3)_2^-$. Thus, m/z +46-containing particles were not classified as amine-containing particles." has been added in the lines 216-220.

"The nitrate-rich amine-containing particles exhibited spectral features different from those observed in the ECOC and BB spectra. Only a few carbonaceous fragments were observed. In summer, the nitrate-rich amine-containing particles contained abundant sea-salt markers such as $m/z$s 23 $[Na]^+$, 62 $[Na_2O]^+$, and 63$[Na_2OH]^+$ in the positive mass spectrum and $m/z$s -93 $[NaCl_2]^-$ and -147 $[Na(NO_3)_2]^-$ in the negative mass spectrum. Chloride signal was not detected due to the depletion of chloride and enrichment of nitrate in the sea-salt particle aging process. In summer, 48-h backward trajectories showed that 60 % of air masses arose from marine areas (Figure S3) and were partly associated with marine aerosols. A small peak of $m/z$ 46 $[(CH_3)_2NH_2]^+$ was found in the nitrate-rich amine-containing particle spectra in summer, which likely arose from DMA produced by marine phytoplankton. The backward trajectories and mass spectra of the nitrate-rich particles indicate that marine sources may contribute to the amine distribution in the PRD region during summer, although the amine-containing particles appeared to have been aged during transportation. In winter, air masses were transported largely from urban areas like Guangzhou and Foshan (Figure S3) and brought more anthropogenic pollutants to the sampling site. Hence, the sea-salt markers at $m/z$s -93 $[NaCl_2]^-$ and -147 $[Na(NO_3)_2]^-$ were not observed in the winter negative mass spectrum. Instead, the $m/z$ 56 $[Fe]^+$ ion was identified, and, because no dust source marker ion signals (such as $Ca^+$, $CaO^+$, and $SiO_3^-$) were found, we speculate that iron arose mainly from industrial emissions. The nitrate-rich amine-containing particles may have resulted from direct industrial emissions or reactions between gaseous amines and particles from industrial emissions. Lastly, the observed nitric acid signal ($m/z$ -125 $[HNO_3NO_3]^-$) indicated strong particle acidity in the nitrate-rich amine-containing particles in winter." have been removed from the manuscript.

3. The authors must be very careful when referring to the particle mixing state. It appears that they confuse the abundances of different particles with the abundances of different chemicals in the same particle. For instance, in Line 314, do they imply that

nitrate and sulfate were present in the same particles and were elevated during summer or that sulfate- and nitrate-containing particles had a high occurrence during summer? Similarly, if I am interpreting the text and Figure 6 correctly, the figure caption should not be the 'mixing state', but the fraction of particles containing different components.

**Response**: The old Figure 6 caption and related expressions were confusing and should be revised as the abundances of ammonium-, nitrate- and sulfate-containing particles instead of "mixing state". In order to demonstrate the abundances of ammonium-, nitrate- and sulfate-containing amine particles in total amine-containing particles, the old Figure 6 has been changed to Table 3, and the related discussions were revised as follows:

Table 3. The abundance of ammonium-, nitrate- and sulfate-containing amine particles in total amine-containing particles.

| Marker ions | Summer | Winter |
|---|---|---|
| $^{18}NH_4^+$ | 8% | 54% |
| $^{62}NO_3^-$ | 43% | 69% |
| $^{97}HSO_4^-$ | 91% | 94% |

The marker ions of $^{18}NH_4^+$, $^{62}NO_3^-$ and $^{97}HSO_4^-$ were chosen to represent ammonium, nitrate and sulfate.

"To investigate the seasonal mixing states of amines with sulfate, nitrate, and ammonium (SNA), the relative abundances of SNA-containing amine particles are shown in Figure 6. The color scale represents the percentage contribution of SNA-containing amine particles to total amine particles. ECOC and BB amine-containing particles were both found to be internally mixed with sulfate throughout the sampling period. Only a small percentage of nitrate-rich amine-containing particles were mixed with sulfate during summer, but this percentage increased to 53 % during winter. Amine particles containing nitrate accounted for 39 % and 59 % of the ECOC and BB particles in summer, respectively, and 68 % and 79 % in winter. The internal mixing state of sulfate and nitrate with amines in single particles suggests the possible formation of aminium sulfate and nitrate salts. Only 6.7 % of the total amine-containing particles contained ammonium in summer, while percentage increased dramatically to 55 % in winter, indicating an ammonium-poor state in summer and an ammonium-rich state in winter." have been revised to "To investigate the aging state of amine-containing particles, the abundances of sulfate-, nitrate-, and ammonium-containing amine particles are shown in Table 3. More than 90% of amine-containing particles were found to be internally mixed with sulfate throughout the sampling period. The abundance of nitrate in amine particles increased from 43% in summer to 69% in winter. The high abundances of sulfate and nitrate in amine-containing particles suggest the possible formation of aminium sulfate and nitrate salts. Interestingly, only 8% of amine-containing particles mixed with ammonium ($NH_4^+$) in summer, while the percentage increased dramatically to 54 % in winter, indicating a relatively $NH_4^+$-poor state in summer and

an NH$_4^+$-rich state in winter." in lines 299-308.

4. As written, it appears that the authors do not treat the charge balance (ion equivalency) correctly when calculating the relative acidity. One cannot simply add the peaks of nitrate and sulfate because the former corresponds to monoprotic and the latter to diprotic acid, respectively. Also, are peak intensities proportional to actual abundancies of chemical species?

**Response**: The relative acidity ratio used in this work is not an equivalent result of the calculation of mass concentrations of anions and cations. It is only used in the studies of single particles mass spectrometer to compare the relative particle acidity of different particles groups. Several related field studies have used this method to discuss the variation of particle acidity (Denkenberger et al., 2007;Pratt et al., 2009;Yao et al., 2011;Huang et al., 2013;Cheng et al., 2017), and Huang et al. (2013) have compared the actual particle acidity calculated from inorganic ions (MARGA data) and relative acidity ratio obtained from single particle mass spectrometer (ATOFMS). The comparison result is as follows:

[Figure]

This graph is from the field study of Huang et al. (2013). The right inset graph shows the linear regression between ATOFMS particle acidity and MARGA particle acidity. The robust correlation between them indicates a feasible estimation of particle acidity through the peak areas of sulfate, nitrate and ammonium obtained from ATOFMS.

We have added the related discussion in the manuscript to state this issue. "According to the field studies of ATOFMS and SPAMS, it is difficult to accurately determine the number concentration of ambient particles using SPAMS alone due to the size-dependent transmission efficiencies of particles through aerodynamic lens and composition dependent matrix effect (Gross et al., 2000;Pratt and Prather, 2012). Thus, the particle counts and size distributions presented in this work should be interpreted as semi-quantitative and serve as a basis of comparison analysis (Healy et

al., 2012).” have been added in lines 185-191.

“Huang et al. (2013) obtained a robust correlation ($r^2$=0.82) between the particle acidity calculated from inorganic ions obtained from MARGA and relative acidity ratio obtained from single particle mass spectrometer, allowing us to use $R_a$ for comparison of particle acidity(Huang et al., 2013).” have been added in lines 350-354 .

The peak area of a chemical species provides a semi-quantitative result of its actual abundance in single particles, which has been specifically discussed in the response to “Comment 2(2)” in detail.

5. Figure 5 is problematic. Does ‘size’ refer to the radius or diameter? If the y-axis is the particle count, then the plotted curves cannot be size distributions. A size distribution is expressed as dCount/dSize, but shown are apparently counts for different size bins. Those must be presented are individual points or bars, not a continuous curve. What is the bin size?

**Response**: We have revised the Figure 5 and related discussions in the manuscript as you suggested. The “size” refers to the diameter of particles, and the y-axis has been changed from “count”to “dC/dlogDp” in Figure 5. The size bin measured by SPAMS is 0.1 μm.

The old Figure 5:

[Figure]

Figure 5. Size distributions of the three types of amine-containing particles in (a) summer and (b) winter.

The new Figure 5:

[Figure]

Figure 5. Unscaled size-resolved number distributions of total amine-containing particles and amine particles containing three marker ions of $^{59}(CH_3)_3N^+$, $^{74}(C_2H_5)_2NH_2^+$, and $^{86}(C_2H_5)_2NCH_2^+$ in summer and winter in Heshan.

"Size-resolved number distributions are shown in Figure 5 for the three types of amine-containing particles. Both ECOC and BB amine-containing particles exhibited unimodal distributions in the submicron mode and had a broad distribution from 0.4 to 1.0 μm in both summer and winter, which may have resulted from amine condensation on and reaction with fine mode particles from anthropogenic emissions. Interestingly, in summer, 37 % of the nitrate-rich amine-containing particles were submicron in size, while 63 % were supermicron; this is reasonable, as the majority of nitrate-rich amine-containing particles were associated with sea-salt particles from marine sources. Healy et al. (2015) also reported large amounts of supermicron amine-containing particles internally mixed with sea-salt particles on the island of Corsica, France." have been revised to "The unscaled size-resolved number distributions of total amine-containing particles and amine particles containing three marker ions of $^{59}(CH_3)_3N^+$, $^{74}(C_2H_5)_2NH_2^+$, and $^{86}(C_2H_5)_2NCH_2^+$ are shown in Figure 5. The amine-containing particles exhibited unimodal distributions in the submicron mode from 0.4 to 1.5 μm in both summer and winter, which may have resulted from gaseous amine condensation on and/or reaction with fine mode particles from anthropogenic emissions. Although amine-containing particles peaked at the size range of 0.5-0.7 μm in both summer and winter, a broader size range of amine-containing particles was observed in winter, which may be due to more complex anthropogenic emission sources of primary particles in winter. The $^{74}(C_2H_5)_2NH_2^+$-containing particles showed similar variation patterns as total amine-containing particles both in summer and winter. However, $^{59}(CH_3)_3N^+$- and $^{86}(C_2H_5)_2NCH_2^+$-containing particles showed less distinct peaks in winter." in lines 285-297.

6. What is the shape of the particle transmission function of the aerodynamic lens?

Have plots shown in Figure 5 been corrected for the size-dependent particle transmission? Frankly, I do not expect the abundance of amine-containing particles to taper off at the smaller sizes. In fact, an opposite should be true. It takes significantly less time to enrich the smaller particles with amines through the substitution reaction than to enrich the larger particles.

**Response**: The size efficiency and hit rate for particles at each size during the measurement by SPAMS are as follows:

[Figure]

Because we didn't have SMPS to measure the ambient particles, size distribution of amine-containing particles was not scaled, thus, we called it "unscaled size distribution". The available size information of particles from SPAMS was used to compare its seasonal difference.

The amines detected in this work were observed in particles with size ranging from 0.4 to 1 μm, which is due to two reasons. Firstly, the SPAMS data is unscaled and many small particles (<300 nm) were undetected during the sampling. Secondly, according to the studies of Sauerwein and Chan, the uptake of dimethylamine (DMA) by acidic particles was severely influenced by the extent of neutralization and the particle phase state. In fully neutralized droplets at 50 % RH, the limited availability of $H_3O^+$ ions for acid-base reactions led to a partial displacement of $NH_4^+$ by the stronger base DMA. At uptake equilibrium, molar ratio of $DMAH/NH_4^+$ in the particles was four times higher than the gas-phase $DMA/NH_3$ concentration ratio, indicating an enrichment of DMA in acidic particles through the displacement of $NH_4^+$ by DMA. In this work the high ambient RH (72 ± 13 %) and acidic particles containing abundant sulfate and nitrate in summer were similar to the experimental conditions used in Sauerwein and Chan (2017). So it is possible that larger particles, with higher degree of neutralization, contain more amines.

7. L175 and everywhere: replace 'm/zs' with 'm/z' L175-180: I suggest placing this information in a table L188-191: provide a reference to the processing method

**Response**: All the "m/zs" have been revised to "m/z" in the manuscript. The marker ions chosen for alkylamines and assignments have been listed in Table 1 as follows.:

Table 1. Marker ions chosen for the amine-containing particles

| Marker ion | Alkylamine assignment |
|---|---|
| $^{59}(CH_3)_3N^+$ | Trimethylamine (TMA) |
| $^{74}(C_2H_5)_2NH_2^+$ | Diethylamine (DEA) |
| $^{86}(C_2H_5)_2NCH_2^+$ | DEA, TEA, DPA |
| $^{101}(C_2H_5)_3N^+$ | Triethylamine (TEA) |
| $^{102}(C_3H_7)_2NH_2^+$ | Dipropylamine (DPA) |
| $^{143}(C_3H_7)_3N^+$ | Tripropylamine (TPA) |

Because the classification rules for ECOC, BB and nitrate-rich type particles are inappropriate, so we removed the description of ART-2a method. "Amine-containing particles were subsequently clustered using the adaptive resonance theory (ART-2a) neural network algorithm with a vigilance factor of 0.75, a learning rate of 0.05, and a maximum of 20 iterations." have been removed from the manuscript.

**References:**

Angelino, S., Suess, D. T., and Prather, K. A.: Formation of aerosol particles from reactions of secondary and tertiary alkylamines: Characterization by aerosol time-of-flight mass spectrometry, Environmental Science & Technology, 35, 3130-3138, Doi 10.1021/Es0015444, 2001.

Bein, K. J., Zhao, Y. J., Pekney, N. J., Davidson, C. I., Johnston, M. V., and Wexler, A. S.: Identification of sources of atmospheric PM at the Pittsburgh Supersite - Part II: Quantitative comparisons of single particle, particle number, and particle mass measurements, Atmospheric Environment, 40, S424-S444, 10.1016/j.atmosenv.2006.01.064, 2006.

Bhave, P. V., Allen, J. O., Morrical, B. D., Fergenson, D. P., Cass, G. R., and Prather, K. A.: A field-based approach for determining ATOFMS instrument sensitivities to ammonium and nitrate, Environmental science & technology, 36, 4868-4879, 2002.

Chan, L. P., and Chan, C. K.: Role of the Aerosol Phase State in Ammonia/Amines Exchange Reactions, Environmental Science & Technology, 47, 5755-5762, 10.1021/es4004685, 2013.

Cheng, C., Li, M., Chan, C. K., Tong, H., Chen, C., Chen, D., Wu, D., Li, L., Wu, C., Cheng, P., Gao, W., Huang, Z., Li, X., Zhang, Z., Fu, Z., Bi, Y., and Zhou, Z.: Mixing state of oxalic acid containing particles in the rural area of Pearl River Delta, China: implications for the formation mechanism of oxalic acid, Atmospheric Chemistry and Physics, 17, 9519-9533, 10.5194/acp-17-9519-2017, 2017.

Chu, Y., and Chan, C. K.: Reactive Uptake of Dimethylamine by Ammonium Sulfate and Ammonium Sulfate–Sucrose Mixed Particles, The Journal of Physical Chemistry A, 121, 206-215, 10.1021/acs.jpca.6b10692, 2016.

Dall'Osto, M., and Harrison, R. M.: Chemical characterisation of single airborne particles in Athens (Greece) by ATOFMS, Atmospheric Environment, 40,

7614-7631, 10.1016/j.atmosenv.2006.06.053, 2006.

Dall'Osto, M., Harrison, R. M., Beddows, D. C., Freney, E. J., Heal, M. R., and Donovan, R. J.: Single-particle detection efficiencies of aerosol time-of-flight mass spectrometry during the North Atlantic marine boundary layer experiment, Environmental science & technology, 40, 5029-5035, 2006.

Denkenberger, K. A., Moffet, R. C., Holecek, J. C., Rebotier, T. P., and Prather, K. A.: Real-time, single-particle measurements of oligomers in aged ambient aerosol particles, Environmental Science & Technology, 41, 5439-5446, 10.1021/es070329l, 2007.

Gaston, C. J., Furutani, H., Guazzotti, S. A., Coffee, K. R., Bates, T. S., Quinn, P. K., Aluwihare, L. I., Mitchell, B. G., and Prather, K. A.: Unique ocean‐derived particles serve as a proxy for changes in ocean chemistry, Journal of Geophysical Research: Atmospheres (1984–2012), 116, 10.1029/2010JD015289, 2011.

Glagolenko, S., and Phares, D. J.: Single-particle analysis of ultrafine aerosol in College Station, Texas, Journal of Geophysical Research-Atmospheres, 109, 10.1029/2004jd004621, 2004.

Gross, D. S., Galli, M. E., Silva, P. J., and Prather, K. A.: Relative sensitivity factors for alkali metal and ammonium cations in single particle aerosol time-of-flight mass spectra, Anal. Chem., 72, 416-422, Doi 10.1021/Ac990434g, 2000.

Guazzotti, S. A., Whiteaker, J. R., Suess, D., Coffee, K. R., and Prather, K. A.: Real-time measurements of the chemical composition of size-resolved particles during a Santa Ana wind episode, California USA, Atmospheric Environment, 35, 3229-3240, 2001.

Healy, R., Sciare, J., Poulain, L., Kamili, K., Merkel, M., Müller, T., Wiedensohler, A., Eckhardt, S., Stohl, A., and Sarda-Estève, R.: Sources and mixing state of size-resolved elemental carbon particles in a European megacity: Paris, Atmospheric Chemistry and Physics, 12, 1681-1700, 2012.

Healy, R. M., Evans, G. J., Murphy, M., Sierau, B., Arndt, J., McGillicuddy, E., O'Connor, I. P., Sodeau, J. R., and Wenger, J. C.: Single-particle speciation of alkylamines in ambient aerosol at five European sites, Analytical and Bioanalytical Chemistry, 407, 5899-5909, 10.1007/s00216-014-8092-1, 2015.

Huang, Y., Chen, H., Wang, L., Yang, X., and Chen, J.: Single particle analysis of amines in ambient aerosol in Shanghai, Environmental Chemistry, 9, 202-210, 10.1071/EN11145, 2012.

Huang, Y., Li, L., Li, J., Wang, X., Chen, H., Chen, J., Yang, X., Gross, D. S., Wang, H., Qiao, L., and Chen, C.: A case study of the highly time-resolved evolution of aerosol chemical and optical properties in urban Shanghai, China, Atmospheric Chemistry and Physics, 13, 3931-3944, 10.5194/acp-13-3931-2013, 2013.

Kane, D. B., and Johnston, M. V.: Enhancing the detection of sulfate particles for laser ablation aerosol mass spectrometry, Anal. Chem., 73, 5365-5369, 10.1021/ac010569s, 2001.

Lake, D. A., Tolocka, M. P., Johnston, M. V., and Wexler, A. S.: The character of single particle sulfate in Baltimore, Atmospheric Environment, 38, 5311-5320, 10.1016/j.atmosenv.2004.02.067, 2004.

Liu, D. Y., Wenzel, R. J., and Prather, K. A.: Aerosol time‑of‑flight mass spectrometry during the Atlanta Supersite Experiment: 1. Measurements, Journal of Geophysical Research: Atmospheres (1984 – 2012), 108, 10.1029/2001JD001562, 2003.

Mansoori, B. A., Johnston, M. V., and Wexler, A. S.: Quantitation of ionic species in single microdroplets by online laser desorption/ionization, Anal. Chem., 66, 3681-3687, 1994.

Phares, D. J., Rhoads, K. P., Johnston, M. V., and Wexler, A. S.: Size-resolved ultrafine particle composition analysis - 2. Houston, Journal of Geophysical Research-Atmospheres, 108, 10.1029/2001jd001212, 2003.

Pratt, K. A., Hatch, L. E., and Prather, K. A.: Seasonal Volatility Dependence of Ambient Particle Phase Amines, Environmental Science & Technology, 43, 5276-5281, 10.1021/es803189n, 2009.

Pratt, K. A., and Prather, K. A.: Mass spectrometry of atmospheric aerosols—Recent developments and applications. Part II: On‑line mass spectrometry techniques, Mass Spectrom Rev, 31, 17-48, 2012.

Qin, X. Y., Bhave, P. V., and Prather, K. A.: Comparison of two methods for obtaining quantitative mass concentrations from aerosol time-of-flight mass spectrometry measurements, Anal. Chem., 78, 6169-6178, 10.1021/ac060395q, 2006.

Sauerwein, M., and Chan, C. K.: Heterogeneous uptake of ammonia and dimethylamine into sulfuric and oxalic acid particles, Atmos. Chem. Phys., 17, 6323-6339, 10.5194/acp-17-6323-2017, 2017.

Snyder, D. C., Schauer, J. J., Gross, D. S., and Turner, J. R.: Estimating the contribution of point sources to atmospheric metals using single-particle mass spectrometry, Atmospheric Environment, 43, 4033-4042, 2009.

Spencer, M. T., and Prather, K. A.: Using ATOFMS to determine OC/EC mass fractions in particles, Aerosol Science and Technology, 40, 585-594, 2006.

Sullivan, R. C., Moore, M. J. K., Petters, M. D., Kreidenweis, S. M., Roberts, G. C., and Prather, K. A.: Timescale for hygroscopic conversion of calcite mineral particles through heterogeneous reaction with nitric acid, Phys Chem Chem Phys, 11, 7826-7837, 10.1039/B904217b, 2009.

Wenzel, R. J., Liu, D. Y., Edgerton, E. S., and Prather, K. A.: Aerosol time-of-flight mass spectrometry during the Atlanta Supersite Experiment: 2. Scaling procedures, Journal of Geophysical Research-Atmospheres, 108, 10.1029/2001jd001563, 2003.

Yao, X. H., Rehbein, P. J. G., Lee, C. J., Evans, G. J., Corbin, J., and Jeong, C. H.: A study on the extent of neutralization of sulphate aerosol through laboratory and field experiments using an ATOFMS and a GPIC, Atmospheric Environment, 45, 6251-6256, 10.1016/j.atmosenv.2011.06.061, 2011.

Zauscher, M. D., Wang, Y., Moore, M. J. K., Gaston, C. J., and Prather, K. A.: Air Quality Impact and Physicochemical Aging of Biomass Burning Aerosols during the 2007 San Diego Wildfires, Environmental Science & Technology, 47, 7633-7643, 10.1021/es4004137, 2013.

Zhang, G., Bi, X., Chan, L. Y., Li, L., Wang, X., Feng, J., Sheng, G., Fu, J., Li, M.,

and Zhou, Z.: Enhanced trimethylamine-containing particles during fog events detected by single particle aerosol mass spectrometry in urban Guangzhou, China, Atmospheric Environment, 55, 121-126, 2012.

---

## Author Comment (AC3) · 22 May 2018

**Response to the comments of Anonymous Referee #3**

[Atmospheric Chemistry and Physics, MS ID: acp-2018-53]
Title: Characteristics and mixing state of amine-containing particles at a rural site in the Pearl River Delta, China.

**General comments:**
This paper uses single-particle mass spectrometry to characterize particle-phase amines observed in a rural part of the Pearl River Delta in China. The analysis in this paper builds upon previous observations of amines to examine the types of amines observed and the role of acidity on the partitioning of particle phase amines using single particle mass spectrometry. Before the paper can be considered for publication, more information is needed about which amine markers were observed on the different particle types as I have reservations about the interpretation of the spectra that must be adequately addressed.

**Response**: Thank you for your comments. We have revised the whole manuscript according to your comments. The classification criteria for BB and nitrate-rich types of particles in the manuscript were inappropriate due to the misinterpretation of several key markers. Thus, we have removed all the description and discussions related with three types of amine-containing particles. Three major amine markers were selected, and the characteristics of these specific amine marker containing particles were investigated. The seasonal different mixing state of amines and ammonium may be influenced by the seasonal variation of source strength of ammonium. Thus, we have added the discussion of ammonium-containing particles in the manuscript. The detailed responses to the comments are itemized below. We appreciate these valuable and helpful comments for improving our paper, as well as the important guiding significance to our researches.

Anything about our paper, please feel free to contact me at limei2007@163.com

Best regards!
Sincerely yours
Mei Li
May 22, 2018

**Specific comments and point by point responses:**

**Major Comments:**

I have several major concerns about this work that need to be addressed:

1. For the nitrate-rich particles observed in summer, m/z +46 is claimed to be an amine peak. However, several single-particle studies note the presence of m/z +46 in sea salt spectra due to the presence of 46Na2+ (e.g., Guazzotti et al., 2001; Gard et al., 1998; Gaston et al., 2011). Was m/z +46 the only "amine" ion peak noted in this particle type? If so, then this ion peak is likely a marker for sodium and not an amine. The text would need to be adjusted and conclusions about marine biogenic amines would need to be removed.

**Response**: We have reconsidered the classification criteria for the nitrate-rich particles which contained many sea-salt markers. Ion of m/z +46 could be $^{46}Na_2^+$ except as amine marker of $^{46}(CH_3)_2NH_2^+$, because no other amine markers were found and nitrate-rich particles were enriched with sodium salts like $^{62}Na_2O^+$, $^{81}Na_2Cl^+$ and $^{147}Na(NO_3)_2^-$, so we removed nitrate-rich particles from amine-containing particles, and all the related discussions were removed too.

"In this work the m/z +46-containing particles had no other amine markers as listed above, besides, these particles were enriched with sodium salts like $^{62}Na_2O^+$, $^{81}Na_2Cl^+$ and $^{147}Na(NO_3)_2^-$. Thus, m/z +46-containing particles were not classified as amine-containing particles." have been added in the lines 216-220.

"The nitrate-rich amine-containing particles exhibited spectral features different from those observed in the ECOC and BB spectra. Only a few carbonaceous fragments were observed. In summer, the nitrate-rich amine-containing particles contained abundant sea-salt markers such as $m/z$s 23 $[Na]^+$, 62 $[Na_2O]^+$, and 63$[Na_2OH]^+$ in the positive mass spectrum and $m/z$s -93 $[NaCl_2]^-$ and -147 $[Na(NO_3)_2]^-$ in the negative mass spectrum. Chloride signal was not detected due to the depletion of chloride and enrichment of nitrate in the sea-salt particle aging process. In summer, 48-h backward trajectories showed that 60 % of air masses arose from marine areas (Figure S3) and were partly associated with marine aerosols. A small peak of $m/z$ 46 $[(CH_3)_2NH_2]^+$ was found in the nitrate-rich amine-containing particle spectra in summer, which likely arose from DMA produced by marine phytoplankton. The backward trajectories and mass spectra of the nitrate-rich particles indicate that marine sources may contribute to the amine distribution in the PRD region during summer, although the amine-containing particles appeared to have been aged during transportation. In winter, air masses were transported largely from urban areas like Guangzhou and Foshan (Figure S3) and brought more anthropogenic pollutants to the sampling site. Hence, the sea-salt markers at $m/z$s -93 $[NaCl_2]^-$ and -147 $[Na(NO_3)_2]^-$ were not observed in the winter negative mass spectrum. Instead, the $m/z$ 56 $[Fe]^+$ ion was identified, and, because no dust source marker ion signals (such as $Ca^+$, $CaO^+$, and $SiO_3^-$) were found, we speculate that iron arose mainly from industrial emissions. The nitrate-rich amine-containing particles may have resulted from direct industrial emissions or reactions between gaseous amines and particles from industrial emissions. Lastly, the observed nitric acid signal ($m/z$ -125

[HNO$_3$NO$_3$]$^-$) indicated strong particle acidity in the nitrate-rich amine-containing particles in winter." have been removed from the manuscript.

"The nitrate-rich amine-containing particles were mixed with abundant sea-salt markers in summer, indicating a possible association between the amine emission source and marine phytoplankton." have been removed from the abstract.

"In summer, the nitrate-rich amine-containing particles were mixed with abundant sea-salt markers, indicating an association between amines and marine phytoplankton emissions." have been removed from the conclusion.

2. I have a similar concern for m/z +74 observed in biomass burning particles. This ion peak is also associated with KCl (e.g., 74KCl+) and may not be indicative of an amine. I suggest that the authors review Zauscher et al., 2013, which does show evidence of biomass burning particles containing amine markers.

**Response**: According to the studies of Zauscher et al. (2013), biomass burning particles were detected with strong signals of potassium chloride ($^{113}$K$_2$Cl$^+$), potassium nitrate ($^{140}$K$_2$NO$_3$$^+$)and potassium sulfate ($^{213}$K$_3$SO$_4$$^+$), so we believe the definition of BB type particles used in this work was inappropriate. Thus, we have removed BB type particles from amine-containing particles, and all the related discussions were removed too. The marker ion of m/z +74 could be $^{74}$KCl$^+$ except as $^{74}$(C$_2$H$_5$)$_2$NH$_2$$^+$, however, no other significant biomass burning markers like $^{113}$K$_2$Cl$^+$, $^{140}$K$_2$NO$_3$$^+$ and $^{213}$K$_3$SO$_4$$^+$ were found in the amine-containing particles, so the presence of m/z +74 in amine-containing particles has a very low possibility to be $^{74}$KCl$^+$.

[revised manuscript text omitted]

3. The authors need to clarify if one amine is seen on the different particle types or multiple markers. I also suggest adding a figure showing the temporal trends for each amine marker.

**Response**: Revision made. We reanalyzed the campaign data as you suggested, and the temporal trends of each amine marker-containing particles are shown in Figure 2.

"Marker ions of $^{59}(CH_3)_3N^+$, $^{74}(C_2H_5)_2NH_2^+$, $^{86}(C_2H_5)_2NCH_2^+$ were detected as the most abundant amines species during the sampling period, so particles containing each marker ion were selected to investigate the possible sources and characteristics of amine-containing particles." have been added in the lines 206-209.

"Among the three markers we considered, the most abundant amine marker was $^{74}(C_2H_5)_2NH_2^+$, which was detected in 90% and 86% of amine-containing particles in summer and winter (Table 2), followed by $^{59}(CH_3)_3N^+$ and $^{86}(C_2H_5)_2NCH_2^+$ which were detected in less than 10% of amine-containing particles during sampling period. The amine particles containing $^{74}(C_2H_5)_2NH_2^+$ and $^{86}(C_2H_5)_2NCH_2^+$ both exhibited similar variation pattern with total amine-containing particles suggesting a similar emission source of $^{74}(C_2H_5)_2NH_2^+$ and $^{86}(C_2H_5)_2NCH_2^+$ (Figure 2). The temporal trend of $^{59}(CH_3)_3N^+$-containing particles were different from those of $^{74}(C_2H_5)_2NH_2^+$ and $^{86}(C_2H_5)_2NCH_2^+$; and the two sudden episodes of $^{59}(CH_3)_3N^+$ occurred from 27 to 29 July in summer were possibly due to the special emission sources of trimethylamine (TMA)." have been added in the lines 245-256.

"Among the three markers we considered, the most abundant amine marker was $^{74}(C_2H_5)_2NH_2^+$, which was detected in 90% and 86% of amine-containing particles in summer and winter, followed by amine marker ions of $^{59}(CH_3)_3N^+$, and $^{86}(C_2H_5)_2NCH_2^+$ which were detected in less than 10% of amine-containing particles during sampling period." have been added in the abstract in lines 51-55.

"Both seasons were dominated by amine marker of $^{74}(C_2H_5)_2NH_2^+$ in 90% and

86% of amine-containing particles in summer and winter, respectively. Amine markers of $^{59}(CH_3)_3N^+$ and $^{86}(C_2H_5)_2NCH_2^+$ were detected in 4.5% and 5.5% of amine-containing particles in summer, while their percentages both increased two times in winter. The amine particles contained $^{74}(C_2H_5)_2NH_2^+$ and $^{86}(C_2H_5)_2NCH_2^+$ both exhibited similar variation pattern with total amine-containing particles suggesting a similar emission source of $^{74}(C_2H_5)_2NH_2^+$ and $^{86}(C_2H_5)_2NCH_2^+$, while the $^{59}(CH_3)_3N^+$-containing particles showed different temporal trends, and two sudden increase episodes of $^{59}(CH_3)_3N^+$ in summer was possibly due to the special sources of trimethylamine." have been added in the conclusion in lines 403-412.

[Figure]

Figure 2. Temporal variations of relative humidity (RH), temperature (T), total amine-containing particles, and three major marker ions-containing amine particles ($^{59}(CH_3)_3N^+$, $^{74}(C_2H_5)_2NH_2^+$, $^{86}(C_2H_5)_2NCH_2^+$) in Heshan, China during sampling periods.

4. The spectra in Figure 4 need to clearly show all of the amine markers observed on the different particle types.

   **Response**: The amine markers of $^{30}CH_3NH^+$, $^{59}(CH_3)_3N^+$, $^{74}(C_2H_5)_2NH_2^+$, $^{86}(C_2H_5)_2NCH_2^+$ have been labeled in the mass spectra of amine-containing particles in Figure 4. The other amine markers of $^{101}(C_2H_5)_3N^+$, $^{102}(C_3H_7)_2NH_2^+$, and $^{143}(C_3H_7)_3N^+$ were found with a much lower peak intensity, and the number count of amine particles contained these markers were less than 1%. So $^{101}(C_2H_5)_3N^+$, $^{102}(C_3H_7)_2NH_2^+$, and $^{143}(C_3H_7)_3N^+$ were not labeled in the mass spectra.

   The old Figure 4:

[Figure]

Figure 4. Average ion mass spectra of ECOC, BB, and nitrate-rich amine-containing particles in (a) summer and (b) winter.

The new Figure 4:

Figure 4. Average ion mass spectra of amine-containing particles in summer and winter.

**Specific Comments:**

Abstract:

1. It would be useful if the authors specified the percentage of amine containing particles that also contained sulfate and/or nitrate.

**Response**: As you suggested, we have added "More than 90% of amine-containing particles were found to be internally mixed with sulfate throughout the sampling period, while the percentage of amine particles containing nitrate increased from 43% in summer to 69% in winter. Robust correlations between the peak intensities of amines and the sum of nitrate and sulfate were observed, suggesting the possible formation of aminium sulfate and nitrate salts." in the abstract in lines 57-62.

2. Lines 54-56: This sentence should be removed. If m/z +46 is the only marker on these aged sea salt particles, then it is likely due to 46Na2+

**Response**: Revision made. "The nitrate-rich amine-containing particles were mixed with abundant sea-salt markers in summer, indicating a possible association between the amine emission source and marine phytoplankton." have been removed from the abstract.

3. Lines 60-61: Is a 9% difference in RH enough to strongly affect the uptake of amines? Could it be that the strength of the ammonium source has a seasonal variation?

**Response**: As you and other reviewers suggested, we have investigated the seasonal distributions of total ammonium-containing particles and the abundance of ammonium in amine-containing particles. The results suggest that the low abundance of ammonium in amine-containing particles in summer is mainly due to the low emission source of ammonia. In addition, according to the study of Sauerwein and Chan, the co-uptake of dimethylamine (DMA) and ammonia ($NH_3$) by sulfuric acid particles at 50% RH led to particle-phase dimethylaminium ($DMAH^+$)/ammonium ($NH_4^+$) molar ratio up to four times that of gas-phase DMA/ammonia molar ratio (0.1 and 0.5), suggesting the displacement of $NH_4^+$ by DMA during the uptake process (Sauerwein and Chan, 2017). Thus, it is feasible for the displacement of $NH_4^+$ by DMA when the concentration of amines is one order of magnitude lower than $NH_3$ in gas phase. The ammonium–amine exchange reactions occurred in summer had little connection with the 9% seasonal difference of RH. We have revised the discussions about the possible role of ammonium–amine exchange reactions in amine-containing particles in the manuscript. Besides, the influence of ambient RH and particle acidity on the gas to particle partitioning of ammonia has been discussed in the manuscript. In the abstract "The ammonium-poor state of amine-containing particles in summer may have been caused by the displacement of particle-phase ammonium by amine uptake, which was more efficient in summer at higher ambient RH (72 ± 13 %) than in winter (63 ± 11 %)." have been revised to "The total ammonium-containing particles were investigated and showed a much lower abundance in ambient particles in summer (3.6%) than that in winter (32.6%), which suggests the ammonium-poor

state of amine-containing particles in summer may be related to the lower abundance of ammonia/ammonium in gas and particle phase. In addition, higher abundance of amines in ammonium-containing particles than that of ammonium in amine-containing particles suggests a possible contribution of ammonium–amine exchange reactions to the low abundance of ammonium in amine-containing particles at high ambient RH (72 ±13 %) in summer." in lines 65-73.

Introduction

1. I suggest adding more references on amines and single-particle observations of amines. Please add Zauscher et al., 2013 in order to comment on biomass burning particles containing amines.

  **Response**: "Zauscher et al. (2013) detected strong signals of amine marker ($^{86}(C_2H_5)_2NCH_2^+$) in biomass burning aerosols associated with the increase of ambient relative humidity, indicating the direct emission of amines from biomass burning and the important influence of high RH (>90%) on the partitioning process of amines." have been added in the introduction in lines 120-124.

  Besides, the reference of Angelino et al. (2001) has also been added in the manuscript. "The mixing state and single-particle characteristics of amines have been investigated in laboratory and field environments (Moffet et al., 2008;Silva et al., 2008;Pratt et al., 2009;Huang et al., 2012;Zhang et al., 2012)." have been revised to "The mixing state and single-particle characteristics of amines have been investigated in laboratory and field environments (Angelino et al., 2001;Moffet et al., 2008;Silva et al., 2008;Pratt et al., 2009;Huang et al., 2012;Zhang et al., 2012)." in lines 111-113.

Methods

1. Lines 173-180: Which fragment goes with which amine? Many ATOFMS studies also note the presence of m/z +30, was this ion also observed? m/z +86 and m/z +118 are some of the most prevalent amines observed using ATOFMS, why wasn't m/z +118 searched for?

  **Response**: We have added a table to show the marker ions chosen for amines and assignments as follows:

Table 1. Marker ions chosen for the amine-containing particles

| Marker ion | Alkylamine assignment |
| --- | --- |
| $^{59}(CH_3)_3N^+$ | Trimethylamine (TMA) |
| $^{74}(C_2H_5)_2NH_2^+$ | Diethylamine (DEA) |
| $^{86}(C_2H_5)_2NCH_2^+$ | DEA, TEA, DPA |
| $^{101}(C_2H_5)_3N^+$ | Triethylamine (TEA) |
| $^{102}(C_3H_7)_2NH_2^+$ | Dipropylamine (DPA) |
| $^{143}(C_3H_7)_3N^+$ | Tripropylamine (TPA) |

  The marker ion of $^{30}CH_3NH^+$ was observed with low peak intensity in 20% of amine-containing particle, and all the particles containing $^{30}CH_3NH^+$ had strong signal of $^{74}(C_2H_5)_2NH_2^+$, so the $^{30}CH_3NH^+$-containing particles were not specifically

discussed. We have added the related discussion about $^{30}CH_3NH^+$ in the manuscript. "In this work the peak intensity of $^{30}CH_3NH^+$ was much lower compared with other amine markers, and all the particles containing $^{30}CH_3NH^+$ had strong signal of $^{74}(C_2H_5)_2NH_2^+$, so the $^{30}CH_3NH^+$-containing particles were not specifically discussed." have been added in the lines 211-214.

The ion of m/z +86 was detected in 5.5% and 9% of amine-containing particles in summer and winter separately. The ion of m/z +118 was detected in less than 100 single particles with very low peak intensity, so we didn't put m/z +118 in the searching list.

2. Lines 173-180: citations are needed for the ion peaks listed and the amines that they correspond to.

**Response**: Revision made. "Based on previous studies using ATOFMS and SPAMS instruments (Angelino et al., 2001;Huang et al., 2012;Zhang et al., 2012;Healy et al., 2015), amine-containing particles were characterized by ionic markers, including $m/z$s 46 $[(CH_3)_2NH_2]^+$, 59 $[(CH_3)_3N]^+$, 74 $[(C_2H_5)_2NH_2]^+$, 86 $[(C_2H_5)_2NCH_2]^+$ or $[C_3H_7NHC_2H_4]^+$, 101 $[(C_2H_5)_3N]^+$, 102 $[(C_3H_7)_2NH_2]^+$, 114 $[(C_3H_7)_2NCH_2]^+$, and 143 $[(C_3H_7)_3N]^+$, which correspond to dimethylamine (DMA), trimethylamine (TMA), diethylamine (DEA), triethylamine (TEA), dipropylamine (DPA), and tripropylamine (TPA)." have been revised to "Based on previous studies using ATOFMS and SPAMS instruments (Angelino et al., 2001;Huang et al., 2012;Zhang et al., 2012;Zauscher et al., 2013;Healy et al., 2015), amine-containing particles were characterized by marker ions, including $m/z$ $^{59}(CH_3)_3N^+$, $^{74}(C_2H_5)_2NH_2^+$, $^{86}(C_2H_5)_2NCH_2^+$, $^{101}(C_2H_5)_3N^+$, $^{102}(C_3H_7)_2NH_2^+$, and $^{143}(C_3H_7)_3N^+$ (Table 1)." in lines 191-195.

Table 1. Marker ions chosen for the amine-containing particles

| Marker ion | Alkylamine assignment |
| --- | --- |
| $^{59}(CH_3)_3N^+$ | Trimethylamine (TMA) |
| $^{74}(C_2H_5)_2NH_2^+$ | Diethylamine (DEA) |
| $^{86}(C_2H_5)_2NCH_2^+$ | DEA, TEA, DPA |
| $^{101}(C_2H_5)_3N^+$ | Triethylamine (TEA) |
| $^{102}(C_3H_7)_2NH_2^+$ | Dipropylamine (DPA) |
| $^{143}(C_3H_7)_3N^+$ | Tripropylamine (TPA) |

3. A search criterion for biomass burning aerosol was m/z -59 and m/z -73, please review Zauscher et al., 2013 for better search criteria for biomass burning aerosols.

**Response**: Thank you for your suggestion. According to the study of Zauscher et al. (2013), biomass burning particles were detected with strong signals of potassium chloride ($^{113}K_2Cl^+$), potassium nitrate ($^{140}K_2NO_3^+$)and potassium sulfate ($^{213}K_3SO_4^+$), so we believe the definition of BB type particles used in this work was inappropriate. Thus, we removed BB type particles from amine-containing particles, and all the related discussions were removed too. The detailed revision has been presented in the

response to Major comment 2.

Results

1. Figure 1 isn't very descriptive. I suggest showing this figure on a map with different sources pointed out so that the reader can see the seasonal impact of different potential sources of amines.

**Response**: Revision made.

The old Figure 1:

[Figure]

Figure 1. Seasonal distributions of amine-containing particle number concentrations associated with wind direction and wind speed in (left) summer and (right) winter.

The new Figure 1:

[Figure]

[Figure]

Figure 1. Spatial distributions of amine-containing particle counts associated with backward trajectories (48 hour) of air masses at 500m levels above the ground during the sampling period: (a) summer (from July 18 to August 1, 2014), (b) winter (from January 27 to February 8, 2015).

"Meteorological conditions, namely wind speed and wind direction, are shown during the sampling period in Figure 1. In summer, high amine-containing particle number concentrations were associated with southwesterly and southeasterly winds at speeds of 3–5 m s$^{-1}$, suggesting that the majority of amine-containing particles came from regional transport. However, in winter, large amounts of amine-containing particles were associated with northwesterly winds at speeds of 0.5–2 m s$^{-1}$, indicating that amine-containing particles were related primarily with local emissions, such as animal husbandry, biomass burning, and vehicle exhaust." have been revised to "Spatial distributions of amine-containing particles associated with backward trajectories (48 hour) of air masses at 500m levels above the ground during the sampling period are shown in Figure 1. Cluster trajectories were calculated by MeteoInfo (Wang, 2014), and the box plots were conducted by Igor Pro-based program Histbox (Wu et al., 2018;Wu and Yu, 2018). In summer, high amine-containing particle counts were associated with air masses of Cluster 3 (41.67%) and Cluster 4 (30.06%) (Figure 1a) from continent and South China Sea separately, suggesting that the majority of amine-containing particles came from anthropogenic and marine sources. However, in winter, large amounts of amine-containing particles were associated with air masses of Cluster 4 (48.08%) (Figure 1b), indicating that amine-containing particles were related primarily with local emissions, such as animal husbandry, biomass burning, and vehicle exhaust.

Anthropogenic emissions from Foshan and Guangzhou may also have contributed, as the sampling site is only 40 km and 56 km from these cities, respectively (Figure S1)." in lines 223-236.

2. Figure 2 has too many traces. I suggest removing wind direction and better separating the amines and the RH. I also suggest showing a temporal trace for each amine marker.

  **Response**: We have revised the Figure 2 as you suggested.
  The old Figure 2:

[Figure]

Figure 2. Temporal variations in amine-containing particles, relative humidity (RH), temperature (T), wind speed (WS), wind direction, and three types of amine particles in Heshan, China during the entire sampling periods. Abbreviations of major particle types: elemental and organic carbon (ECOC); biomass burning (BB).

  The new Figure 2:

[Figure]

Figure 2. Temporal variations of relative humidity (RH), temperature (T), total

amine-containing particles, and amine particles containing three major marker ions of $^{59}(CH_3)_3N^+$, $^{74}(C_2H_5)_2NH_2^+$, $^{86}(C_2H_5)_2NCH_2^+$ in Heshan, China during the entire sampling periods.

3. Figure 3 needs standard error bars. Do different amine peaks show different diurnal trends?

**Response**: We have revised the Figure 3 as you suggested.

The old Figure 3:

[Figure]

Figure 3. Diurnal variations in amine-containing particle number concentrations in summer and winter in Heshan.

The new Figure 3:

[Figure]

Figure 3. Diurnal variations of amine-containing particle counts in summer and winter in Heshan.

"The amine particles containing $^{74}(C_2H_5)_2NH_2^+$ and $^{86}(C_2H_5)_2NCH_2^+$ both exhibited similar variation pattern with total amine-containing particles suggesting a similar emission source of $^{74}(C_2H_5)_2NH_2^+$ and $^{86}(C_2H_5)_2NCH_2^+$ (Figure 2). The

temporal trend of $^{59}(CH_3)_3N^+$-containing particles were different from those of $^{74}(C_2H_5)_2NH_2^+$ and $^{86}(C_2H_5)_2NCH_2^+$; and the two sudden episodes of $^{59}(CH_3)_3N^+$ occurred from 27 to 29 July in summer were possibly due to the special emission sources of trimethylamine (TMA)." have been added in the lines 249-256.

4. Figure 4 needs amine markers clearly labeled on each particle type. I am also surprised to not see m/z +86 and m/z +118 as these are some of the most prevalent amine markers observed by single particle mass spectrometry.

**Response**: Revision made. The amine markers of $^{30}CH_3NH^+$, $^{59}(CH_3)_3N^+$, $^{74}(C_2H_5)_2NH_2^+$, $^{86}(C_2H_5)_2NCH_2^+$ have been labeled in the mass spectra of amine-containing particles in Figure 4. The other amine markers of $^{101}(C_2H_5)_3N^+$, $^{102}(C_3H_7)_2NH_2^+$, and $^{143}(C_3H_7)_3N^+$ were found with a much lower peak intensity, and the number count of amine particles contained these markers were less than 1%. So $^{101}(C_2H_5)_3N^+$, $^{102}(C_3H_7)_2NH_2^+$, and $^{143}(C_3H_7)_3N^+$ were not labeled in the mass spectra. The ion of m/z +118 was detected in less than 100 single particles with very low peak intensity, so m/z +118 was not labeled in the Figure 4.

The new Figure 4:

[Figure]

Figure 4. Average ion mass spectra of amine-containing particles in summer and winter. The color bars represent each peak area corresponding to a specific ion in individual particles.

5. From Figure 4, it appears as though m/z +74 is the only amine marker observed on biomass burning aerosols. m/z +74 is also a marker for KCl (e.g., 74KCl+) as noted in Zauscher et al., 2013 and may be misclassified as an amine in this work. Were any other amine markers observed on this particle type?

**Response**: According to the study of Zauscher et al. (2013), biomass burning particles were detected with strong signals of potassium chloride ($^{113}K_2Cl^+$), potassium nitrate ($^{140}K_2NO_3^+$)and potassium sulfate ($^{213}K_3SO_4^+$), so we believe if m/z +74 was $^{74}KCl^+$ , the significant biomass burning markers listed above should be detected in the particles. Biomass burning markers like $^{113}K_2Cl^+$, $^{140}K_2NO_3^+$ and

$^{213}K_3SO_4^+$ were not found in the amine-containing particles, so the presence of m/z +74 in amine-containing particles has a very low possibility to be $^{74}KCl^+$. In this work the ions of $^{30}CH_3NH^+$ and $^{86}(C_2H_5)_2NCH_2^+$ were also detected in the amine particles containing $^{74}(C_2H_5)_2NH_2^+$.

6. Figure 5 should clearly state that the x-axis is vacuum aerodynamic diameter.

    **Response**: The label of x-axis has been revised as you suggested:

[Figure]

Figure 5. Unscaled size-resolved number distributions of total amine-containing particles and amine particles containing three marker ions of $^{59}(CH_3)_3N^+$, $^{74}(C_2H_5)_2NH_2^+$, and $^{86}(C_2H_5)_2NCH_2^+$ in summer and winter in Heshan.

7. Figure 6 could be a subtraction plot between the two seasons to better illustrate the seasonal difference in the mixing-state of the amine-containing particles.

    **Response**: Indeed, the seasonal different abundance of sulfate, nitrate and ammonium are not easily identified in Figure 6 through the illustration of color squares. Thus, we have put the percentages of amine particles containing sulfate, nitrate and ammonium in Table 3 instead of Figure 6.

Table 3. The abundances of ammonium-, nitrate- and sulfate-containing amine particles in total amine-containing particles.

| Marker ions | Summer | Winter |
|---|---|---|
| $^{18}NH_4^+$ | 8% | 54% |
| $^{62}NO_3^-$ | 43% | 69% |
| $^{97}HSO_4^-$ | 91% | 94% |

The marker ions of $^{18}NH_4^+$, $^{62}NO_3^-$ and $^{97}HSO_4^-$ were chosen to represent ammonium, nitrate and sulfate.

8. The biomass burning spectra and ECOC spectra look the same. I think these should be the same particle type. See Ault et al., 2010 for representative ECOC spectra, which do not contain intense ions at 39K+.

**Response**: As suggested by you, the classification criteria were inappropriate for the BB and nitrate-rich type particles. Thus, we no longer use the art-2a method to classify the amine-containing particles. We have combined the ECOC and BB types of particles together to investigate the mass spectra and mixing state of amine-containing particles. The mass spectra and size distributions of ECOC and BB type particles were replaced by total amine-containing particles, and all the discussions about ECOC and BB type particles were all replaced by total amine-containing particles.

[revised manuscript text omitted]

9. Lines 277-279: chloride is not completely displaced and is detected as 81, 83Na2Cl+ and 93, 95NaCl2-.

**Response**: Thanks for the correction. The depletion of chloride is nitrate-rich particles is worthy of discussion. However, because the m/z +46 was not identified as

amine marker, all the discussions about nitrate-rich type particles have been removed. Although this issue is no longer investigated in this work, the comment from you is still valuable and helpful to our future research.

10. Lines 281-283: I am not convinced that this is an observation of a marine-derived amine. The spectra clearly show aged sea salt particles, which should contain an ion peak at 46Na2+. Therefore, the peak at m/z +46 is likely not an amine.

**Response**: Revision made. We have made an incorrect assignment of m/z +46, and the discussions about the marine contribution to amines have been removed.

"The nitrate-rich amine-containing particles exhibited spectral features different from those observed in the ECOC and BB spectra. Only a few carbonaceous fragments were observed. In summer, the nitrate-rich amine-containing particles contained abundant sea-salt markers such as $m/z$s 23 $[Na]^+$, 62 $[Na_2O]^+$, and 63$[Na_2OH]^+$ in the positive mass spectrum and $m/z$s -93 $[NaCl_2]^-$ and -147 $[Na(NO_3)_2]^-$ in the negative mass spectrum. Chloride signal was not detected due to the depletion of chloride and enrichment of nitrate in the sea-salt particle aging process (Gard et al., 1998). In summer, 48-h backward trajectories showed that 60 % of air masses arose from marine areas (Figure S3) and were partly associated with marine aerosols. A small peak of $m/z$ 46 $[(CH_3)_2NH_2]^+$ was found in the nitrate-rich amine-containing particle spectra in summer, which likely arose from DMA produced by marine phytoplankton (Facchini et al., 2008). The backward trajectories and mass spectra of the nitrate-rich particles indicate that marine sources may contribute to the amine distribution in the PRD region during summer, although the amine-containing particles appeared to have been aged during transportation. In winter, air masses were transported largely from urban areas like Guangzhou and Foshan (Figure S3) and brought more anthropogenic pollutants to the sampling site. Hence, the sea-salt markers at $m/z$s -93 $[NaCl_2]^-$ and -147 $[Na(NO_3)_2]^-$ were not observed in the winter negative mass spectrum. Instead, the $m/z$ 56 $[Fe]^+$ ion was identified, and, because no dust source marker ion signals (such as $Ca^+$, $CaO^+$, and $SiO_3^-$) were found, we speculate that iron arose mainly from industrial emissions. The nitrate-rich amine-containing particles may have resulted from direct industrial emissions or reactions between gaseous amines and particles from industrial emissions. Lastly, the observed nitric acid signal ($m/z$ -125 $[HNO_3NO_3]^-$) indicated strong particle acidity in the nitrate-rich amine-containing particles in winter." have been removed from the manuscript.

11. Lines 333-334: could it be that the source strength of ammonium shows a seasonal cycle? This is a more likely explanation than differences in partitioning caused by a very small change in RH.

**Response**: As suggested by you, we have studied the total ammonium-containing particles during sampling period to investigate the source strength of ammonium. The results show that the low abundance of ammonium in amine-containing particles in summer could be a possible result of low emission source of ammonia.

"The seasonal differences of the mixing state of amines and $NH_4^+$ may be influenced by the seasonal variation of source strength of $NH_4^+$. To investigate the temporal variation and abundance of $NH_4^+$ in total detected single particles, the total $NH_4^+$-containing particles were identified with relative area of $^{18}NH_4^+$ larger than 1%. Using this criterion, 18336 and 235312 of $NH_4^+$-containing particles were detected in summer and winter separately, accounting for 3.6% and 32.6% of the total detected particles. The averaged positive and negative ion mass spectra of $NH_4^+$-containing particles are exhibited in Figure 6. During entire sampling period the $NH_4^+$-containing particles were characterized by abundant hydrocarbon fragments and secondary organic species like $^{43}C_2H_3O^+$ and $^{89}HC_2O_4^-$, as well as strong signals of $^{26}CN^-$, $^{42}CNO^-$, $^{62}NO_3^-$ and $^{97}HSO_4^-$, indicating an aging state of $NH_4^+$-containing particles. Also, 20% of $NH_4^+$-containing particles contained $^{74}(C_2H_5)_2NH_2^+$, which indicates a close connection between $NH_3$ and diethylamine (DEA), possibly due to the similar emission sources. Temporal variations of total amine-containing particles, total ammonium-containing ($NH_4^+$-containing) particles and particles containing both ammonium and amine ($NH_4^+$-amine) are shown in Figure 7. The total $NH_4^+$-containing particles and $NH_4^+$-amine particles were both much lower in summer than in winter. This seasonal difference may be due to the low emission sources of ammonia and preferred partitioning in gas phase in summer. Backward trajectories analysis (Figure 1) showed that in summer the air mass was mainly from south of the sampling site and linked to the marine region with low emission of anthropogenic pollutants. By contrast, in winter, the air mass was mainly from northwest of the sampling site and associated with relatively polluted megacities like Guangzhou and Foshan. RH does not seem to exert a major influence on particulate $NH_4^+$ (Huang et al., 2012), because lower abundance of $NH_4^+$ was observed in summer (RH = 72 ± 13%) than in winter (RH =63 ± 11%)." have been added in the manuscript in lines 309-335.

[Figure]

Figure 6. Mass spectra of total ammonium-containing ($NH_4^+$-containing) particles in summer and winter. The color bars represent each peak area corresponding to a specific fraction in individual particles.

[Figure]

Figure 7. Temporal variations of total amine-containing particles, total ammonium-containing particles and ammonium-containing amine particles ($NH_4^+$-amine) during sampling period in Heshan.

12. The authors are quite redundant about the displacement of ammonium by amines. This should be re-read and redundancies should be reduced.

**Response**: Revision made. As suggested by you, we have investigated the ammonium-containing particles in summer, and the low abundance of ammonium in amine-containing particles in summer is closely associated with the low emission of ammonia. In addition, according to the study of Sauerwein and Chan, the co-uptake of dimethylamine (DMA) and ammonia ($NH_3$) by sulfuric acid particles at 50% RH led to particle-phase dimethylaminium ($DMAH^+$)/ammonium ($NH_4^+$) molar ratio up to four times that of gas-phase DMA/ammonia molar ratio (0.1 and 0.5), suggesting the displacement of $NH_4^+$ by DMA during the uptake process (Sauerwein and Chan, 2017). In this work, the ambient RH and acidic particles containing abundant sulfate and nitrate were similar to the experimental conditions used in Sauerwein and Chan (2017). In summer 8% of amine-containing particles contained $NH_4^+$, while 25% of ammonium-containing particles contained amines (Figure 7). Although the gas-phase concentrations of amines and $NH_3$ are not quantified, higher abundance of amines in ammonium-containing particles than that of ammonium in amine-containing particles suggests a possible ammonium–amine exchange reactions in acidic particles in summer. Thus, we have extended the related discussion about ammonium–amine exchange reactions.

"Although high acidity is favorable for gaseous ammonia partitioning, extremely low ammonium peak areas were found for the amine-containing particles in summer (Figure 7), which may have been caused by the displacement of ammonium in the particle phase by amines to form aminium sulfate and nitrate salts; this displacement depends on RH and the phase of the ammonium salts (Chan and Chan, 2013;Chu and Chan, 2016). The ambient RH in summer (72 ± 13 %) was higher than that in winter (63 ± 11 %). Thus, it is feasible that particles contained more water and

a larger fraction of aqueous ammonium salts in summer than in winter, which facilitated the displacement of ammonium by amines, decreasing the ammonium concentration in the particle phase. Particle-phase organics other than amines and aminium salts also affect amine–ammonia exchange (Chu and Chan, 2016, 2017). However, because the SPAMS alone cannot provide quantitative data on particle-phase organics, this issue will be discussed in a subsequent study." have been revised to "Although high acidity promotes gaseous ammonia partitioning, extremely low ammonium peak areas were found for the amine-containing particles in summer (Figure 8), which may be associated with ammonium–amine exchange reactions in addition to the low emission source of ammonia. The exchange between amine gases and particulate $NH_3$ and/or ammonium highly depends on the RH and particle acidity (Chan and Chan, 2013;Chu and Chan, 2016). According to the study of Sauerwein and Chan, the co-uptake of dimethylamine (DMA) and ammonia ($NH_3$) by sulfuric acid particles at 50% RH led to particle-phase dimethylaminium ($DMAH^+$) to ammonium ($NH_4^+$) molar ratio up to four times that of gas-phase DMA to ammonia molar ratio (0.1 and 0.5), suggesting the displacement of $NH_4^+$ by DMA during the uptake process (Sauerwein and Chan, 2017). In this work, the ambient RH and acidic particles containing abundant sulfate and nitrate were similar to the experimental conditions used in Sauerwein and Chan (2017). In summer 8% of amine-containing particles contained $NH_4^+$, while 25% of ammonium-containing particles contained amines (Figure 7). Although the gas-phase concentrations of amines and $NH_3$ are not quantified, higher abundance of amines in ammonium-containing particles than that of ammonium in amine-containing particles suggests a possible ammonium–amine exchange reactions in acidic particles in summer." in lines 356-373.

In the abstract, "The ammonium-poor state of amine-containing particles in summer may have been caused by the displacement of particle-phase ammonium by amine uptake, which was more efficient in summer at higher ambient RH (72 $\pm$13 %) than in winter (63 $\pm$ 11 %)." have been revised to "In addition, higher abundance of amines in ammonium-containing particles than that of ammonium in amine-containing particles suggests a possible contribution of ammonium–amine exchange reactions to the low abundance of ammonium in amine-containing particles at high ambient RH (72 $\pm$13 %) in summer." in lines 69-73.

In the conclusion, "The ammonium-poor state of the amine-containing particles in summer may have been caused by the displacement of ammonium in the particle phase by amines to form aminium sulfate and nitrate salts." have been revised to "Besides, 8% of amine-containing particles contained ammonium while 25% of ammonium-containing particles contained amines in summer, suggesting a possible contribution of ammonium–amine exchange reactions to the low abundance of ammonium in amine-containing particles at high ambient RH (72 $\pm$13 %) in summer." in lines 424-428.

Conclusions:
1. The authors should comment on which amine markers were most prevalent and on

which particle types. This would be an interesting conclusion that may also tie into different potential amine sources.

[revised manuscript text omitted]

---

## Referee Report (RR1)

**Characteristics and mixing state of amine-containing particles at a rural site in the Pearl River Delta, China**

By Cheng et al.

General:

This paper has been significantly improved from the first version. I have a few comments that should be addressed before publication.

Major Comments:

My main comments are:

1. Figure 8 is a bit problematic. Correlations between sulfate and nitrate on amine-containing particles is used to infer how amines are formed. Sulfate and nitrate should be separated out in this figure in order to determine the formation of different aminium salts. Also, the correlations are quite misleading since amine-containing particles were compared against each other instead of sulfate, nitrate, and amine markers on all particles.
2. The conclusions could be significantly strengthened by moving the commentary on lines 382-398 that links sulfate formation in Chinese haze to mixing state and amines to the conclusions.

Specific Comments:
Introduction
1. I suggest adding one few more reference on amines on lines 89-90. Please add [*Facchini et al.*, 2008].
2. Lines 111-113: Please also add [*Gaston et al.*, 2013; *Qin et al.*, 2012; *Zauscher et al.*, 2013].

Methods
1. Was a silica gel drier used during sampling to reduce particle phase water?
2. Lines 191-195: Please also cite [*Gaston et al.*, 2013; *Qin et al.*, 2012].
3. In table 1, include references for each ion peak.
4. Line 220: Please also add to the end of the sentence "and are likely sea salts". This will help avoid confusion.

Results
1. Line 231: From Figure 1, it looks like the open ocean trajectory has the fewest amines. Perhaps your amines aren't "marine sources" per say but are derived from coastal emissions.
2. Line 232: Cluster 4 for the winter is very stagnant. I would guess that those stagnant conditions also facilitate the partitioning of amines.
3. Line 263-266: What about the role of temperature?
4. Figure 4 needs to have actual m/z values on the ion peaks. For example, instead of $CH_3NH$, show $^{30}CH_3NH^+$.
5. Lines 287-291: was the size distribution for amines any different than the size distribution for all particles? If not, then this figure and discussion is not very important.
6. Line 298: I suggest removing "formation processes" in the title of section 3.3. I am not completely convinced that you can completely deduce this information.

7. Lines 299-308: I suggest comparing your seasonal trends of amines and ammonium to Qin et al., 2012, which also contains observations of ammonium nitrate and amines.
8. I find Figure 7 to be very interesting. What is the $R^2$ between amines and ammonium-rich particles? It looks high in the winter.
9. Lines 333-335: RH has been shown to exert an influence on compounds such as ammonium nitrate. I suggest revising this sentence to reflect that your findings indicate that source seems more important than RH for explaining seasonal trends.
10. Figure 8 just shows a peak area comparison on amine comparing particles. The trends reported in lines 339-341 are misleading because only amine-containing particles were selected so if these particles also had sulfate, the correlation would be high. How do the correlations hold if you compare the ion peak areas for all particles instead?
11. Lines 347-349: The authors should explicitly state that this method was developed by Pratt et al., 2009 for single particle work since the authors use the exact same methodology.
12. Lines 386-398 should be moved to the conclusions section.

Conclusions:

1. I recommend removing the sentence on lines 417-419. Correlation does not equal causation.
2. This section has conclusions but no implications. I recommend removing lines 429-436 and replacing it with lines 386-398.

Technical Comments:

1. Line 49: change "count" to "counts"
2. Line 214: change "ion of m/z" to "an ion peak at m/z"
3. Line 250: change "similar variation pattern" to "a similar pattern"
4. Line 255 and 412: remove "special"
5. Line 258: change "count" to "counts"
6. Line 299: change "aging" to "mixing"
7. Line 316: change "During entire" to "During the entire"
8. Line 319: change "aging" to "mixing"
9. Line 379: remove "Besides"
10. Line 386: change "hence the mixing state…" to "As pointed out in Pratt et al., 2009 and in this work, the mixing state…"

References:

Facchini, M. C., S. Decesari, M. Rinaldi, C. Carbone, E. Finessi, M. Mircea, S. Fuzzi, F. Moretti, E. Tagliavini, D. Ceburnis, and C. D. O'Dowd (2008), Important source of marine secondary organic aerosol from biogenic amines, *Environmental Science & Technology*, *42*(24), 9116-9121.

Gaston, C. J., P. K. Quinn, T. S. Bates, J. B. Gilman, D. M. Bon, W. C. Kuster, and K. A. Prather (2013), The impact of shipping, agricultural, and urban emissions on single particle chemistry observed aboard the R/V Atlantis during CalNex, *Journal of Geophysical Research-Atmospheres*, *118*, doi:10.1002/jgrd.50427.

Qin, X., K. A. Pratt, L. G. Shields, S. M. Toner, and K. A. Prather (2012), Seasonal comparisons of single-particle chemical mixing state in Riverside, CA, *Atmospheric Environment*, *59*, 587-596.

Zauscher, M. D., Y. Wang, M. J. K. Moore, C. J. Gaston, and K. A. Prather (2013), Air quality impact and physicochemical aging of biomass burning aerosols during the 2007 San Diego wildfires, *Environmental Science & Technology*, DOI: 10.1021/es4004137.

---

## Author Response (AR2)

**Response letter**

[Atmospheric Chemistry and Physics, MS ID: acp-2018-53]
Title: Characteristics and mixing state of amine-containing particles at a rural site in the Pearl River Delta, China.

**Comments to the Authors:**
Both referees of your revised manuscript are in favor of publication but have a few technical and minor revision requests. Please consider the referees' comments in revising your manuscript before publication of your work proceeds.

**Response**: Thank you for your comments and contribution to our manuscript. We have revised the whole manuscript according to the reports from two anonymous referees, and the point-by-point responses have been listed below. We appreciate these valuable and helpful comments and suggestions to our work and enable it to meet the high quality of the journal Atmos. Chem. Phys. All the revisions made in the manuscript have been highlighted with blue color. The marked-up manuscript has been uploaded.

Anything about our paper, please feel free to contact me at limei2007@163.com

Best regards!
Sincerely yours

Mei Li
June 17, 2018

**Response to the report #1 from anonymous referee #2**

**Comments:**

The corrections are satisfactory. The authors should use a smooth line instead of bins in Figure 5 because now it presents size distributions.

**Response**: Thank you for your comments. We have revised the Figure 5 according to your suggestion. The vertical lines have been replaced by log-normal fitting curves. The old Figure 5:

[Figure]

Figure 5. Unscaled size-resolved number distributions of total amine-containing particles and amine particles containing three marker ions of $^{59}(CH_3)_3N^+$, $^{74}(C_2H_5)_2NH_2^+$, and $^{86}(C_2H_5)_2NCH_2^+$ in summer and winter in Heshan.

The new Figure 5:

[Figure]

Figure 5. Unscaled size-resolved number distributions of total amine-containing particles and amine particles containing three marker ions of $^{59}(CH_3)_3N^+$, $^{74}(C_2H_5)_2NH_2^+$, and $^{86}(C_2H_5)_2NCH_2^+$ in summer and winter in Heshan.

**Response to the report #2 from anonymous referee #3**

**General comments:**

This paper has been significantly improved from the first version. I have a few comments that should be addressed before publication.

**Response**: Thank you for your comments. We have revised the manuscript according to your comments and suggestions, and the point-by-point responses are listed below.

**Major Comments:**

1. Figure 8 is a bit problematic. Correlations between sulfate and nitrate on amine-containing particles is used to infer how amines are formed. Sulfate and nitrate should be separated out in this figure in order to determine the formation of different aminium salts. Also, the correlations are quite misleading since amine-containing particles were compared against each other instead of sulfate, nitrate, and amine markers on all particles.

**Response**: Thank you for your suggestion. The seasonal trends of the peak areas of sulfate and nitrate have been separated out in new Figure 8, and the related discussions have been revised. The linear regressions in old Figure S3 have also been revised to the linear regressions between amines and sulfate and nitrate separately in new Figure S4. In this work all the particles containing amines have been picked up as amine-containing particles, so no amine markers existed in the rest particles, thus, the correlations between sulfate, nitrate and amines on all particles were actually the correlations between them in amine-containing particles.

"The temporal variations of the peak areas of amines, ammonium, and the sum of sulfate and nitrate in amine-containing particles are shown in Figure 8. The peak areas of amines and the sum of nitrate and sulfate had similar variation patterns both in summer and winter. The linear regression between them showed robust correlations both in summer ($r^2$=0.74) and winter ($r^2$=0.88) (Figure S4), indicating the formation of aminium salts." have been revised to "The temporal variations of the peak areas of amines, ammonium, sulfate and nitrate in amine-containing particles are shown in Figure 8. The peak areas of amines and sulfate had similar variation patterns both in summer and winter, and the linear regression between them showed robust correlations both in summer ($r^2$=0.69) and winter ($r^2$=0.72) (Figure S4), indicating the formation of aminium sulfate salt during the entire sampling period. However, the peak areas of amines and nitrate only exhibited similar trends in winter, and the linear regression between them showed a better correlation in winter ($r^2$=0.78) than in summer ($r^2$=0.52) (Figure S4), suggesting the possible formation of aminium nitrate salt in winter." in lines 348-356.

"The sum of the sulfate and nitrate peak areas had a higher increase rate than the amine peak area from 6 to 8 February, which may have been caused by an increase of ammonium during this period." have been removed from the manuscript.

"Robust correlations between the peak intensities of amines and the sum of nitrate and sulfate were observed, suggesting the possible formation of aminium sulfate and nitrate salts." have been revised to "Robust correlations between the peak intensities of amines and sulfate and nitrate were observed, suggesting the possible

formation of aminium sulfate and nitrate salts." in the abstract in lines 60-62.

The old Figure 8:

[Figure]

Figure 8. Temporal variations in the peak areas of amines, ammonium, and the sum of sulfate and nitrate in amine-containing particles during summer and winter. The relative acidity ratio ($R_a$), which was calculated as the ratio of the total sulfate and nitrate peak areas to the ammonium peak area, is plotted as log($R_a$).

The new Figure 8:

[Figure]

Figure 8. Temporal variations of the peak areas of amines, ammonium, sulfate and nitrate in amine-containing particles during summer and winter. The relative acidity ratio ($R_a$), which was calculated as the ratio of the total sulfate and nitrate peak areas to the ammonium peak area, is plotted as log($R_a$).

The old Figure S3:

[Figure]

Figure S3. The linear regressions between peak area of amines and the peak area of sulfate and nitrate in summer and winter.

The new Figure S4:

[Figure]

Figure S4. The linear regressions between the peak areas of amines and sulfate in summer (a) and winter (c), and the linear regressions between the peak areas of amines and nitrate in summer (b) and winter (d) in amine-containing particles.

2. The conclusions could be significantly strengthened by moving the commentary on lines 382-398 that links sulfate formation in Chinese haze to mixing state and amines to the conclusions.

**Response**: Revision made. "In addition, the presence of aminium salts affects the water activities and osmotic coefficients of aqueous solutions, which may influence the calculation of pH using aerosol thermodynamic models (Sauerwein et al., 2015). Furthermore, it should be noted that the measured pH of bulk ambient aerosols may not be representative of the actual single particle acidity. As pointed out in Pratt et al.

(2009) and in this work, the mixing state of aerosols should be considered in order to comprehensively estimate the aerosol pH. Several recent studies have reported a 'missing' source of sulfate produced from the oxidation of $SO_2$ by $NO_2$ during haze episodes with high ambient relative humidity in northern China, and the neutralization of particulate ammonium is a key factor in this formation mechanism (Cheng et al., 2016;Wang et al., 2016). Our study reveals that amines have a potential influence on particle acidity, which could also impact this sulfate formation process during haze episodes. In order to discuss the potential role of amines in this sulfate formation pathway, real-time concentrations of amines, ammonium, sulfate, nitrate, and their precursors must be available. The results of this study suggest that amine chemistry involving particle acidity and mixing state with sulfate, nitrate and ammonium may have an important role in the aging process of particles in regions with high concentration of amines." have been moved to the conclusions in lines 425-441.

**Specific Comments:**
**Introduction**
1. I suggest adding one few more reference on amines on lines 89-90. Please add [Facchini et al., 2008].
**Response**: Revision made. [Facchini et al., 2008] has been cited and added in line 90.

2. Lines 111-113: Please also add [Gaston et al., 2013; Qin et al., 2012; Zauscher et al., 2013].
**Response**: Revision made. [Gaston et al., 2013; Qin et al., 2012; Zauscher et al., 2013] have been cited and added in lines 112-114.

**Methods**
1. Was a silica gel drier used during sampling to reduce particle phase water?
**Response**: Yes, a silica gel drier was used to dry the ambient particles before arriving at the SPAMS. "The SPAMS was installed at the top of the main building, and aerosols were introduced to the SPAMS through a 2.5 m copper tube." have been revised to "The SPAMS was installed at the top of the main building, and aerosols were introduced to the SPAMS through a 2.5 m copper tube and a silica gel drier." in lines 156-157.

2. Lines 191-195: Please also cite [Gaston et al., 2013; Qin et al., 2012].
**Response**: Revision made. [Gaston et al., 2013; Qin et al., 2012] have been cited and added in lines 192-193.

3. In table 1, include references for each ion peak.
**Response**: Revision made. The new Table 1:

Table 1. Marker ions chosen for the amine-containing particles

| Marker ion | Alkylamine assignment |
|---|---|
| $^{59}(CH_3)_3N^+$ | Trimethylamine (TMA)[a] |
| $^{74}(C_2H_5)_2NH_2^+$ | Diethylamine (DEA)[b] |
| $^{86}(C_2H_5)_2NCH_2^+$ | DEA, TEA, DPA[c] |
| $^{101}(C_2H_5)_3N^+$ | Triethylamine (TEA)[d] |
| $^{102}(C_3H_7)_2NH_2^+$ | Dipropylamine (DPA)[e] |
| $^{143}(C_3H_7)_3N^+$ | Tripropylamine (TPA)[f] |

References are as follows: [a]Zhang et al., 2012, Gaston et al., 2013; [b]Angelino et al., 2001; [c]Huang et al., 2012, Zauscher et al., 2013, Qin et al., 2012; [d]Gaston et al., 2013; [e]Pratt et al., 2009; [f]Healy et al., 2015.

4. Line 220: Please also add to the end of the sentence "and are likely sea salts". This will help avoid confusion.
**Response**: Revision made. "and are likely sea salts" have been added in lines 220-221.

**Results**
1. Line 231: From Figure 1, it looks like the open ocean trajectory has the fewest amines. Perhaps your amines aren't "marine sources" per say but are derived from coastal emissions.
**Response**: Revision made. "suggesting that the majority of amine-containing particles came from anthropogenic and marine sources" has been revised to "suggesting that the majority of amine-containing particles came from anthropogenic sources and coastal emissions" in lines 231-232.

2. Line 232: Cluster 4 for the winter is very stagnant. I would guess that those stagnant conditions also facilitate the partitioning of amines.
**Response**: Revision made. "Besides, the stagnant meteorological conditions associated with Cluster 4 also facilitated the partitioning of amines from gas to particle phase in winter." have been added in lines 238-239.

3. Line 263-266: What about the role of temperature?
**Response**: Thank you for your suggestion. No significant diurnal temperature differences were found both in summer and winter, and the related discussions "Although lower temperature facilitates the partitioning of gaseous amines into the particulate phase (Huang et al., 2012), no significant temperature differences were found both in summer (day: 32±1.1 ℃; night: 27.5±1.1 ℃) and winter (day: 15±1.4 ℃; night: 13±0.8 ℃), which suggests a minor influence of temperature on the diurnal pattern of amine-containing particles." have been added in lines 268-273.

4. Figure 4 needs to have actual m/z values on the ion peaks. For example, instead of CH3NH, show 30CH3NH+.

**Response**: We have revised the Figure 4 according to your suggestion. Besides, the old Figure 6 has also been revised due to the same reason.

The new Figure 4:

[Figure]

Figure 4. Average ion mass spectra of amine-containing particles in summer and winter. The color bars represent each peak area corresponding to a specific ion in individual particles.

The new Figure 6:

[Figure]

Figure 6. Mass spectra of total ammonium-containing ($NH_4^+$-containing) particles in summer and winter. The color bars represent each peak area corresponding to a specific fraction in individual particles.

5. Lines 287-291: was the size distribution for amines any different than the size distribution for all particles? If not, then this figure and discussion is not very important.

**Response**: The unscaled size-resolved number distributions of total detected particles and amine-containing particles in summer and winter are shown as follows. In

summer, the total detected particles and amine-containing particles exhibited similar unimodal distributions in the submicron mode from 0.4 to 1.5 μm. However, in winter, the peak of total detected particles shifted to a larger size range from 0.6 to 0.9 μm compared with the size range of amine-containing particles from 0.5 to 0.8 μm. Therefore, the size distributions of amine-containing particles were different from the size distributions of total detected particles. Thus, we believe the related discussions about amine-containing particles are useful to infer the sources and possible formation process of amines in particles.

[Figure]

Figure R1. Unscaled size-resolved number distributions of total detected particles and amine-containing particles in summer and winter in Heshan.

6. Line 298: I suggest removing "formation processes" in the title of section 3.3. I am not completely convinced that you can completely deduce this information.
**Response**: Revision made. "3.3 Mixing state and formation processes of amine-containing particles" has been revised to "3.3 Mixing state of amine-containing particles" in line 305.

7. Lines 299-308: I suggest comparing your seasonal trends of amines and ammonium to Qin et al., 2012, which also contains observations of ammonium nitrate and amines.
**Response**: Revision made. "The high abundances of sulfate and nitrate in amine-containing particles suggest the possible formation of aminium sulfate and nitrate salts." have been revised to "The internal mixing state of amines with sulfate

and nitrate had also been found in Qin et al. (2012), which reported that the amine-rich particles consisted of 18±10% amines by mass in the form of aminium sulfate and nitrate salts in summer in Riverside, California. In this work, the high abundances of sulfate and nitrate in amine-containing particles suggest the possible formation of aminium sulfate and nitrate salts." in lines 310-315.

8. I find Figure 7 to be very interesting. What is the R2 between amines and ammonium-rich particles? It looks high in the winter.
**Response**: The linear regressions between amine-containing particles and ammonium-containing particles in summer and winter are shown in the Supplement as Figure S3:

[Figure]

Figure S3. The linear regressions between amine-containing particles and ammonium-containing particles in summer and winter.

"and the $NH_4^+$-containing particles and amine-containing particles showed a robust linear correlation in winter ($r^2$=0.63) (Figure S3)." have been added in lines 337-338.

9. Lines 333-335: RH has been shown to exert an influence on compounds such as ammonium nitrate. I suggest revising this sentence to reflect that your findings indicate that source seems more important than RH for explaining seasonal trends.
**Response**: Thank you for your suggestion. We have revised "RH does not seem to exert a major influence on particulate $NH_4^+$ (Huang et al., 2012), because lower abundance of $NH_4^+$ was observed in summer (RH = 72 ± 13%) than in winter (RH =63 ± 11%)." to "In this work the lower abundance of $NH_4^+$ was observed in summer (RH = 72 ± 13%) than in winter (RH =63 ± 11%), suggesting a more important influence of sources than RH on the seasonal trends of $NH_4^+$-containing particles." in lines 344-347.

10. Figure 8 just shows a peak area comparison on amine comparing particles. The trends reported in lines 339-341 are misleading because only amine-containing particles were selected so if these particles also had sulfate, the correlation would be high. How do the correlations hold if you compare the ion peak areas for all particles

instead?

**Response**: In this work all the particles containing amines have been picked up as amine-containing particles, so no amine markers existed in the rest particles, thus, the correlations between sulfate, nitrate and amines on all particles were actually the correlations between them in amine-containing particles. As we have made the revisions in the response to Major Comment 1, the seasonal trends of the peak areas of sulfate and nitrate have been separated out in new Figure 8 to illustrate the formation of aminium sulfate and nitrate salts. In addition, the linear regressions in old Figure S3 have also been revised to the linear regressions between amines and sulfate and nitrate separately in new Figure S4.

"The temporal variations of the peak areas of amines, ammonium, and the sum of sulfate and nitrate in amine-containing particles are shown in Figure 8. The peak areas of amines and the sum of nitrate and sulfate had similar variation patterns both in summer and winter. The linear regression between them showed robust correlations both in summer ($r^2$=0.74) and winter ($r^2$=0.88) (Figure S4), indicating the formation of aminium salts." have been revised to "The temporal variations of the peak areas of amines, ammonium, sulfate and nitrate in amine-containing particles are shown in Figure 8. The peak areas of amines and sulfate had similar variation patterns both in summer and winter, and the linear regression between them showed robust correlations both in summer ($r^2$=0.69) and winter ($r^2$=0.72) (Figure S4), indicating the formation of aminium sulfate salt during the entire sampling period. However, the peak areas of amines and nitrate only exhibited similar trends in winter, and the linear regression between them showed a better correlation in winter ($r^2$=0.78) than in summer ($r^2$=0.52) (Figure S4), suggesting the possible formation of aminium nitrate salt in winter." in lines 348-356.

The new Figure 8:

[Figure]

Figure 8. Temporal variations of the peak areas of amines, ammonium, sulfate and nitrate in amine-containing particles during summer and winter. The relative acidity ratio ($R_a$), which was calculated as the ratio of the total sulfate and nitrate peak areas to the ammonium peak area, is plotted as log($R_a$).

The new Figure S4:

[Figure]

Figure S4. The linear regressions between the peak areas of amines and sulfate in summer (a) and winter (c), and the linear regressions between the peak areas of amines and nitrate in summer (b) and winter (d) in amine-containing particles.

11. Lines 347-349: The authors should explicitly state that this method was developed by Pratt et al., 2009 for single particle work since the authors use the exact same methodology.
**Response**: Revision made. "In this work the particle acidity of amine-containing particles is represented by the relative acidity ratio ($R_a$), which is defined as the ratio of the sum of the sulfate and nitrate peak areas divided by the ammonium peak area (Denkenberger et al., 2007;Pratt et al., 2009;Cheng et al., 2017)." have been revised to "In this work the particle acidity of amine-containing particles is represented by the relative acidity ratio ($R_a$), which is developed by Denkenberger et al. (2007) and Pratt et al. (2009), defined as the ratio of the sum of the sulfate and nitrate peak areas divided by the ammonium peak area (Denkenberger et al., 2007;Pratt et al., 2009;Cheng et al., 2017)." in lines 360-363.

12. Lines 386-398 should be moved to the conclusions section.
**Response**: Revision made. "In addition, the presence of aminium salts affects the water activities and osmotic coefficients of aqueous solutions, which may influence the calculation of pH using aerosol thermodynamic models (Sauerwein et al., 2015). Furthermore, it should be noted that the measured pH of bulk ambient aerosols may not be representative of the actual single particle acidity. As pointed out in Pratt et al. (2009) and in this work, the mixing state of aerosols should be considered in order to comprehensively estimate the aerosol pH. Several recent studies have reported a

'missing' source of sulfate produced from the oxidation of $SO_2$ by $NO_2$ during haze episodes with high ambient relative humidity in northern China, and the neutralization of particulate ammonium is a key factor in this formation mechanism (Cheng et al., 2016;Wang et al., 2016). Our study reveals that amines have a potential influence on particle acidity, which could also impact this sulfate formation process during haze episodes. In order to discuss the potential role of amines in this sulfate formation pathway, real-time concentrations of amines, ammonium, sulfate, nitrate, and their precursors must be available. The results of this study suggest that amine chemistry involving particle acidity and mixing state with sulfate, nitrate and ammonium may have an important role in the aging process of particles in regions with high concentration of amines." have been moved to the conclusion in lines 425-441.

**Conclusions:**

1. I recommend removing the sentence on lines 417-419. Correlation does not equal causation.

**Response**: Revision made. "Robust correlations between the peak intensities of amines and the sum of nitrate and sulfate suggested the possible formation of aminium sulfate and nitrate salts." have been removed from the conclusion.

2. This section has conclusions but no implications. I recommend removing lines 429-436 and replacing it with lines 386-398.

**Response**: Revision made. "In order to estimate the particle acidity, the relative acidity ratio ($R_a$), defined as the ratio of the sum of the sulfate and nitrate peak areas divided by the ammonium peak area, was calculated and showed higher values in summer (326 ± 326) than (31 ± 13) in winter, suggesting the amine-containing particles were more acidic in summer than in winter. However, after including amines along with the ammonium in the acidity calculation, the new $R_a'$ values showed no distinct seasonal change (summer: 11 ± 4; winter: 10 ± 2), suggesting that it is reasonable to consider amines when estimating particle acidity." have been replaced by "In addition, the presence of aminium salts affects the water activities and osmotic coefficients of aqueous solutions, which may influence the calculation of pH using aerosol thermodynamic models (Sauerwein et al., 2015). Furthermore, it should be noted that the measured pH of bulk ambient aerosols may not be representative of the actual single particle acidity. As pointed out in Pratt et al. (2009) and in this work, the mixing state of aerosols should be considered in order to comprehensively estimate the aerosol pH. Several recent studies have reported a 'missing' source of sulfate produced from the oxidation of $SO_2$ by $NO_2$ during haze episodes with high ambient relative humidity in northern China, and the neutralization of particulate ammonium is a key factor in this formation mechanism (Cheng et al., 2016;Wang et al., 2016). Our study reveals that amines have a potential influence on particle acidity, which could also impact this sulfate formation process during haze episodes. In order to discuss the potential role of amines in this sulfate formation pathway, real-time concentrations of amines, ammonium, sulfate, nitrate, and their precursors must be available. The results of this study suggest that amine chemistry involving particle acidity and

mixing state with sulfate, nitrate and ammonium may have an important role in the aging process of particles in regions with high concentration of amines." in lines 425-441.

**Technical Comments:**
1. Line 49: change "count" to "counts"
**Response**: Revision made. "Although the increase of amine-containing particle count mostly occurred at night" has been revised to "Although the increase of amine-containing particle counts mostly occurred at night" in lines 48-49.

2. Line 214: change "ion of m/z" to "an ion peak at m/z"
**Response**: Revision made. "Ion of m/z +46 was detected in the ambient single particles" has been revised to "An ion peak at m/z +46 was detected in the ambient single particles" in lines 214-215.

3. Line 250: change "similar variation pattern" to "a similar pattern"
**Response**: Revision made. "The amine particles containing $^{74}(C_2H_5)_2NH_2^+$ and $^{86}(C_2H_5)_2NCH_2^+$ both exhibited similar variation pattern with total amine-containing particles" has been revised to "The amine particles containing $^{74}(C_2H_5)_2NH_2^+$ and $^{86}(C_2H_5)_2NCH_2^+$ both exhibited a similar pattern with total amine-containing particles" in lines 252-254.

4. Line 255 and 412: remove "special"
**Response**: Revision made. "the special emission sources of trimethylamine (TMA)." has been revised to "the emission sources of trimethylamine (TMA)." in line 258. "the special sources of trimethylamine." has been revised to "the emission sources of trimethylamine." in line 410.

5. Line 258: change "count" to "counts"
**Response**: Revision made. "both showed higher count at night" has been revised to "both showed higher counts at night" in line 260.

6. Line 299: change "aging" to "mixing"
**Response**: Revision made. "To investigate the aging state of amine-containing particles" has been revised to "To investigate the mixing state of amine-containing particles" in line 306.

7. Line 316: change "During entire" to "During the entire"
**Response**: Revision made. "During entire sampling period" has been revised to "During the entire sampling period" in line 326.

8. Line 319: change "aging" to "mixing"
**Response**: Revision made. "indicating an aging state of $NH_4^+$-containing particles" has been revised to "indicating an mixing state of $NH_4^+$-containing particles" in lines

329-330.

9. Line 379: remove "Besides"
**Response**: Revision made. "Besides" has been removed in line 393.

10. Line 386: change "hence the mixing state…" to "As pointed out in Pratt et al., 2009 and in this work, the mixing state…"
**Response**: Revision made. "Hence, the mixing state of aerosols should be considered in order to comprehensively estimate the aerosol pH" has been revised to "As pointed out in Pratt et al. (2009) and in this work, the mixing state of aerosols should be considered in order to comprehensively estimate the aerosol pH." in lines 429-431.

**References:**

Cheng, C., Li, M., Chan, C. K., Tong, H., Chen, C., Chen, D., Wu, D., Li, L., Wu, C., Cheng, P., Gao, W., Huang, Z., Li, X., Zhang, Z., Fu, Z., Bi, Y., and Zhou, Z.: Mixing state of oxalic acid containing particles in the rural area of Pearl River Delta, China: implications for the formation mechanism of oxalic acid, Atmospheric Chemistry and Physics, 17, 9519-9533, 10.5194/acp-17-9519-2017, 2017.

Cheng, Y., Zheng, G., Wei, C., Mu, Q., Zheng, B., Wang, Z., Gao, M., Zhang, Q., He, K., Carmichael, G., Pöschl, U., and Su, H.: Reactive nitrogen chemistry in aerosol water as a source of sulfate during haze events in China, Science Advances, 2, 10.1126/sciadv.1601530, 2016.

Denkenberger, K. A., Moffet, R. C., Holecek, J. C., Rebotier, T. P., and Prather, K. A.: Real-time, single-particle measurements of oligomers in aged ambient aerosol particles, Environmental Science & Technology, 41, 5439-5446, 10.1021/es070329l, 2007.

Huang, Y., Chen, H., Wang, L., Yang, X., and Chen, J.: Single particle analysis of amines in ambient aerosol in Shanghai, Environmental Chemistry, 9, 202-210, 10.1071/EN11145, 2012.

Pratt, K. A., Hatch, L. E., and Prather, K. A.: Seasonal Volatility Dependence of Ambient Particle Phase Amines, Environmental Science & Technology, 43, 5276-5281, 10.1021/es803189n, 2009.

Qin, X. Y., Pratt, K. A., Shields, L. G., Toner, S. M., and Prather, K. A.: Seasonal comparisons of single-particle chemical mixing state in Riverside, CA, Atmospheric Environment, 59, 587-596, 10.1016/j.atmosenv.2012.05.032, 2012.

Sauerwein, M., Clegg, S. L., and Chan, C. K.: Water Activities and Osmotic Coefficients of Aqueous Solutions of Five Alkylaminium Sulfates and Their Mixtures with H2SO4 at 25(o)C, Aerosol Science and Technology, 49, 566-579, 10.1080/02786826.2015.1043045, 2015.

Wang, G., Zhang, R., Gomez, M. E., Yang, L., Levy Zamora, M., Hu, M., Lin, Y., Peng, J., Guo, S., Meng, J., Li, J., Cheng, C., Hu, T., Ren, Y., Wang, Y., Gao, J., Cao, J., An, Z., Zhou, W., Li, G., Wang, J., Tian, P., Marrero-Ortiz, W., Secrest, J.,

Du, Z., Zheng, J., Shang, D., Zeng, L., Shao, M., Wang, W., Huang, Y., Wang, Y., Zhu, Y., Li, Y., Hu, J., Pan, B., Cai, L., Cheng, Y., Ji, Y., Zhang, F., Rosenfeld, D., Liss, P. S., Duce, R. A., Kolb, C. E., and Molina, M. J.: Persistent sulfate formation from London Fog to Chinese haze, Proceedings of the National Academy of Sciences, 113, 13630-13635, 10.1073/pnas.1616540113, 2016.

---

## Author Response (AR3)

**Response letter**

[Atmospheric Chemistry and Physics, MS ID: acp-2018-53]
Title: Characteristics and mixing state of amine-containing particles at a rural site in the Pearl River Delta, China.

**Comments to the Authors:**
I am happy to accept your manuscript for publication in Atmospheric Chemistry and Physics.
Technical corrections:
line 411: Please change "count" to "counts".

**Response**: Thank you for your comments and contribution to our manuscript. We have revised the manuscript according to your comment, and the point-by-point response has been listed below. We appreciate your valuable and helpful comment and suggestion to our work and enable it to meet the high quality of the journal Atmos. Chem. Phys.

Anything about our paper, please feel free to contact me at limei2007@163.com

Best regards!
Sincerely yours

Mei Li
June 21, 2018

**Response to the specific comment**

**Technical corrections:** line 411: Please change "count" to "counts".
**Response**: Revision made. "Although the increase of amine-containing particle count mostly occurred at night" has been revised to "Although the increase of amine-containing particle counts mostly occurred at night" in line 411.